# GOLPH3 and GOLPH3L maintain Golgi localization of LYSET and a functional mannose 6-phosphate transport pathway

Berit K Brauer [ID][1,9], Zilei Chen [ID][2,9], Felix Beirow [ID][1], Jiaran Li[3], Daniel Meisinger[1], Emanuela Capriotti[4], Michaela Schweizer[5], Lea Wagner [ID][1], Jascha Wienberg [ID][1], Laura Hobohm [ID][1], Lukas Blume [ID][1,8], Wenjie Qiao[6], Yoshiki Narimatsu[7], Jan E Carette[6], Henrik Clausen [ID][7], Dominic Winter [ID][3], Thomas Braulke[4], Sabrina Jabs [ID][2✉] & Matthias Voss [ID][1✉]

## Abstract

Glycosylation, which plays an important role in modifying lipids and sorting of proteins, is regulated by asymmetric intra-Golgi distribution and SPPL3-mediated cleavage of Golgi enzymes. We found that cells lacking LYSET/TMEM251, a retention factor for Golgi *N*-acetylglucosamine-1-phosphotransferase (GNPT), display SPPL3-dependent hypersecretion of the Golgi membrane protein B4GALT5. We demonstrate that in wild-type cells B4GALT5 is tagged with mannose 6-phosphate (M6P), a sorting tag typical of soluble lysosomal hydrolases. Hence, M6P-tagging of B4GALT5 may represent a novel degradative lysosomal pathway. We also observed B4GALT5 hypersecretion and prominent destabilization of LYSET–GNPT complexes, impaired M6P-tagging, and disturbed maturation and trafficking of lysosomal enzymes in multiple human cell lines lacking the COPI adaptors GOLPH3 and GOLPH3L. Mechanistically, we identified LYSET as a novel, atypical client of GOLPH3/GOLPH3L. Thus, by ensuring the *cis*-Golgi localization of the LYSET–GNPT complex and maintaining its Golgi polarity, GOLPH3/GOLPH3L is essential for the integrity of the M6P-tagging machinery and homeostasis of lysosomes.

**Keywords** Golgi Apparatus; Lysosomes; Mannose 6-Phophate Tagging; Glycosyltransferase Secretion; Intramembrane Proteolysis
**Subject Categories** Membranes & Trafficking; Organelles

See also: **N Zubkov & S Munro**

## Introduction

The Golgi apparatus harbors an ensemble of close to 200 enzymes that ensure the decoration of proteins and lipids with complex, often branched oligosaccharides (Schjoldager et al, 2020). In multicellular organisms, Golgi glycosylation is fundamentally important for many physiological processes (Ohtsubo and Marth, 2006) and defects in genes implicated in Golgi glycosylation cause severe diseases (Ng and Freeze, 2018). In addition, host-pathogen interactions are often glycosylation-dependent and malignancies are associated with altered glycosylation patterns (Pinho and Reis, 2015; Kremsreiter et al, 2021).

Proteolytic cleavage of Golgi-resident glycosylation enzymes (such as glycosyltransferases) and their subsequent release from the Golgi apparatus is a largely unexplored mechanism regulating cellular Golgi glycosylation (Voss, 2024). Our previous work established that the Golgi-resident intramembrane protease signal peptide peptidase-like 3 (SPPL3) cleaves a substantial number of Golgi type II membrane proteins off their membrane anchor, enabling their release from the Golgi apparatus and secretion (Voss et al, 2014; Kuhn et al, 2015; Hobohm et al, 2022). Underscoring the potency of SPPL3 as a regulator of cellular glycosylation, numerous unbiased cell-based genetic screens have since linked loss of *SPPL3* to overt phenotypes such as glycosylation-dependent tumor immune evasion (Dufva et al, 2020; Jongsma et al, 2021; Heard et al, 2022; Dufva et al, 2023; Zhuang et al, 2024).

Recently, the COPI adaptors GOLPH3 and GOLPH3L were shown to facilitate retrieval of select Golgi type II membrane proteins, helping to maintain their correct intra-Golgi localization (Rizzo et al, 2021; Welch et al, 2021). Also, LYSET (previously called TMEM251) was recently shown to ensure *cis*-Golgi retention of the heterohexameric (α2, β2, γ2) GlcNAc-1-phosphotranferase (GNPT) complex (Pechincha et al, 2022; Richards et al, 2022), encoded by the *GNPTAB* and *GNPTG* genes, respectively (reviewed

[1]Institute of Biochemistry, Kiel University, Kiel, Germany. [2]Institute of Clinical Molecular Biology, Kiel University and University Hospital Schleswig-Holstein, Campus Kiel, Kiel, Germany. [3]Institute for Biochemistry and Molecular Biology, Medical Faculty, Rheinische Friedrich-Wilhelms-University of Bonn, Bonn, Germany. [4]Department of Osteology and Biomechanics, Cell Biology of Rare Diseases, University Medical Center Hamburg-Eppendorf, Hamburg, Germany. [5]Morphology and Electron Microscopy, University Medical Center Hamburg-Eppendorf, Center for Molecular Neurobiology (ZMNH), Hamburg, Germany. [6]Department of Microbiology and Immunology, Stanford University School of Medicine, Stanford, CA, USA. [7]Faculty of Health Sciences, Centre for Glycomics, Department of Cellular and Molecular Medicine, University of Copenhagen, Copenhagen, Denmark. [8]Present address: Institute of Cellular and Integrative Physiology, University Medical Center Hamburg Eppendorf, Hamburg, Germany. [9]These authors contributed equally: Berit K Brauer, Zilei Chen. ✉E-mail: s.jabs@ikmb.uni-kiel.de; mvoss@biochem.uni-kiel.de

in (Braulke et al, 2024)). The GNPTAB precursor protein is activated by proteolytic cleavage in the Golgi apparatus (Marschner et al, 2011). GNPT is a key enzyme involved in mannose 6-phosphate (M6P)-tagging of soluble lysosomal proteins, a prerequisite for their efficient M6P-specific receptor (MPR)-mediated sorting and targeting to lysosomes, whereas lysosomal membrane proteins are targeted to lysosomes via specific sorting motifs in their cytoplasmic tails (Braulke et al, 2024). In LYSET-deficient cells, GNPT is mislocalized and degraded in lysosomes, preventing M6P-tagging of about 70 lysosomal enzymes and resulting in striking lysosomal dysfunctions (Pechincha et al, 2022; Richards et al, 2022). This implies that both the expression and the correct *cis*-Golgi localization of LYSET are crucial to cellular M6P-tagging and thus the biogenesis of lysosomes. Importantly, however, it has remained enigmatic how Golgi localization of LYSET is ensured.

Here, we report that LYSET deficiency results in a profound, SPPL3-dependent hypersecretion of β1,4-galactosyltransferase 5 (referred to by its gene name B4GALT5), a key enzyme in Golgi glycosphingolipid synthesis (D'Angelo et al, 2013), and we found that its unconventional M6P-tagging serves as lysosomal targeting signal of the cleaved B4GALT5. Notably, GOLPH3/GOLPH3L-deficient cell lines also secrete soluble B4GALT5 and in addition display a substantial reduction of LYSET levels, concomitant destablization of GNPT, impaired M6P-tagging and hypersecretion of lysosomal enzymes. Mechanistically, our work suggests that LYSET represents a novel, atypical GOLPH3/GOLPH3L client and that GOLPH3 and GOLPH3L are critically required to maintain steady-state Golgi localization of the LYSET/GNPT complex and a functional M6P-dependent protein transport pathway.

# Results

## Loss of LYSET leads to SPPL3-dependent secretion of B4GALT5

To assess whether loss of LYSET also affects Golgi-localized enzymes apart from GNPT, we re-analyzed our previously reported intracellular proteome and secretome data of *Lyset* KO mouse embryonic fibroblasts (MEF) (Richards et al, 2022) specifically for type II membrane proteins and known SPPL3 substrates. Surprisingly, we found that only the galactosyltransferase B4GALT5 was markedly increased in the *Lyset* KO secretome (Fig. 1A), while secretion of other type II membrane proteins, including the previously reported SPPL3 substrates B4GAT1 (Voss et al, 2014) and GALNT2 (Hobohm et al, 2022) was largely unaffected by *Lyset* deficiency. Immunoblotting of conditioned media from 293FT (Fig. 1B) and HAP1 (Fig. 1C) cells using two distinct B4GALT5-specific monoclonal antibodies confirmed the pronounced B4GALT5 secretion into media of *LYSET* KO cells, whereas no B4GALT5 was detectable in media of parental (WT) cells. The steady-state concentration of B4GALT5 in the enriched membrane fraction was too low for antibody detection. In both 293FT and HAP1 *LYSET* KO cells, secretion of the well-established SPPL3 substrates MGAT5, B4GALT1, B4GAT1, and GALNT2 (Voss et al, 2014; Kuhn et al, 2015; Hobohm et al, 2022) was not altered (Fig. 1B,C), demonstrating that loss of LYSET did not globally influence secretion of known SPPL3 substrates and thus

does not generally affect SPPL3 activity. To determine whether B4GALT5 secretion from *LYSET* KO cells was SPPL3-dependent, we generated *LYSET*- and *SPPL3*-double-deficient 293FT cells using Cas9 genome editing. In three clones, successful targeting of the *SPPL3* locus and loss of SPPL3 protein were confirmed by Sanger sequencing (Fig. 1D) and by immunoblotting (Fig. 1E), respectively. *LYSET/SPPL3* KO 293FT clones displayed markedly reduced secretion of SPPL3 substrates, of which B4GALT1 and MGAT5 concomitantly accumulated in the membrane fraction (Fig. 1F). Importantly, no soluble B4GALT5 could be detected in conditioned media of *LYSET/SPPL3* KO clones, whereas hypersecretion of the soluble lysosomal lipid synthase CLN5 and the protease cathepsin D due to their impaired M6P-tagging was still apparent (Fig. 1G), excluding a generalized secretion defect. This demonstrated that cell-endogenous SPPL3 is required for cleavage of the membrane anchor and subsequent secretion of B4GALT5 from LYSET-deficient 293FT cells.

## B4GALT5 secreted from LYSET-deficient cells lacks M6P residues

*LYSET* KO cells hypersecrete numerous soluble lysosomal proteins due to impaired M6P-tagging that precludes their lysosomal targeting (Pechincha et al, 2022; Richards et al, 2022; Zhang et al, 2022). Therefore, we assessed whether B4GALT5 secretion from LYSET-deficient cells is caused by the lack of M6P moieties and analyzed isogenic *GNPTAB* KO 293FT (Fig. 2A) and HAP1 cells (Fig. 2B). These cells express correctly localized LYSET (Richards et al, 2022), but are unable to modify lysosomal enzymes with M6P-tags due to a loss of the catalytic subunits of GNPT encoded by *GNPTAB*. As expected, *LYSET* and *GNPTAB* KO cell lines displayed cathepsin L hypersecretion (Fig. 2A,B). Importantly, also *GNPTAB*-deficient cells displayed markedly increased secretion of endogenous B4GALT5 resembling the observation made for LYSET-deficient cells. This suggests that B4GALT5 secretion is not specifically caused by the absence of its LYSET-mediated retention but by a lack of M6P formation in these cell lines.

To further substantiate that B4GALT5 is subject to M6P-tagging and MPR-dependent sorting, we treated WT 293FT cells with ammonium chloride. Ammonium chloride prevents the low pH-dependent dissociation of MPR-ligand complexes and leads to occupied MPRs along all transport routes. Thus, all newly synthesized M6P-tagged proteins are secreted and not targeted to lysosomes (Braulke et al, 1987). Following ammonium chloride treatment, both B4GALT5-specific antibodies detected a distinct band in conditioned media of WT 293FT cells (Fig. 2C), which was absent in conditioned media of *B4GALT5* KO cells (Fig. 2D). The lower electrophoretic mobility of B4GALT5 secreted from LYSET-deficient cells is due to complex oligosaccharide modifications instead of M6P-modified high mannose-type oligosaccharides, which was abolished upon deglycosylation using PNGase F (Fig. 2C). Next, we knocked out *SPPL3* in Flp-In T-REx 293 cells and re-introduced either doxycycline-inducible SPPL3 WT or the catalytically inactive D271N mutant (Voss et al, 2014) (Appendix Fig. S1). Using these models, we found that catalytic activity of SPPL3 is indeed required for ammonium chloride-induced secretion of B4GALT5 (Fig. 2E) and B4GALT5 secretion was similarly dependent on endogenous SPPL3 in hTERT RPE-1 cells (Appendix Fig. S2). Finally, affinity purification of exogenously expressed V5-tagged B4GALT5 using an M6P-recognizing single-chain antibody

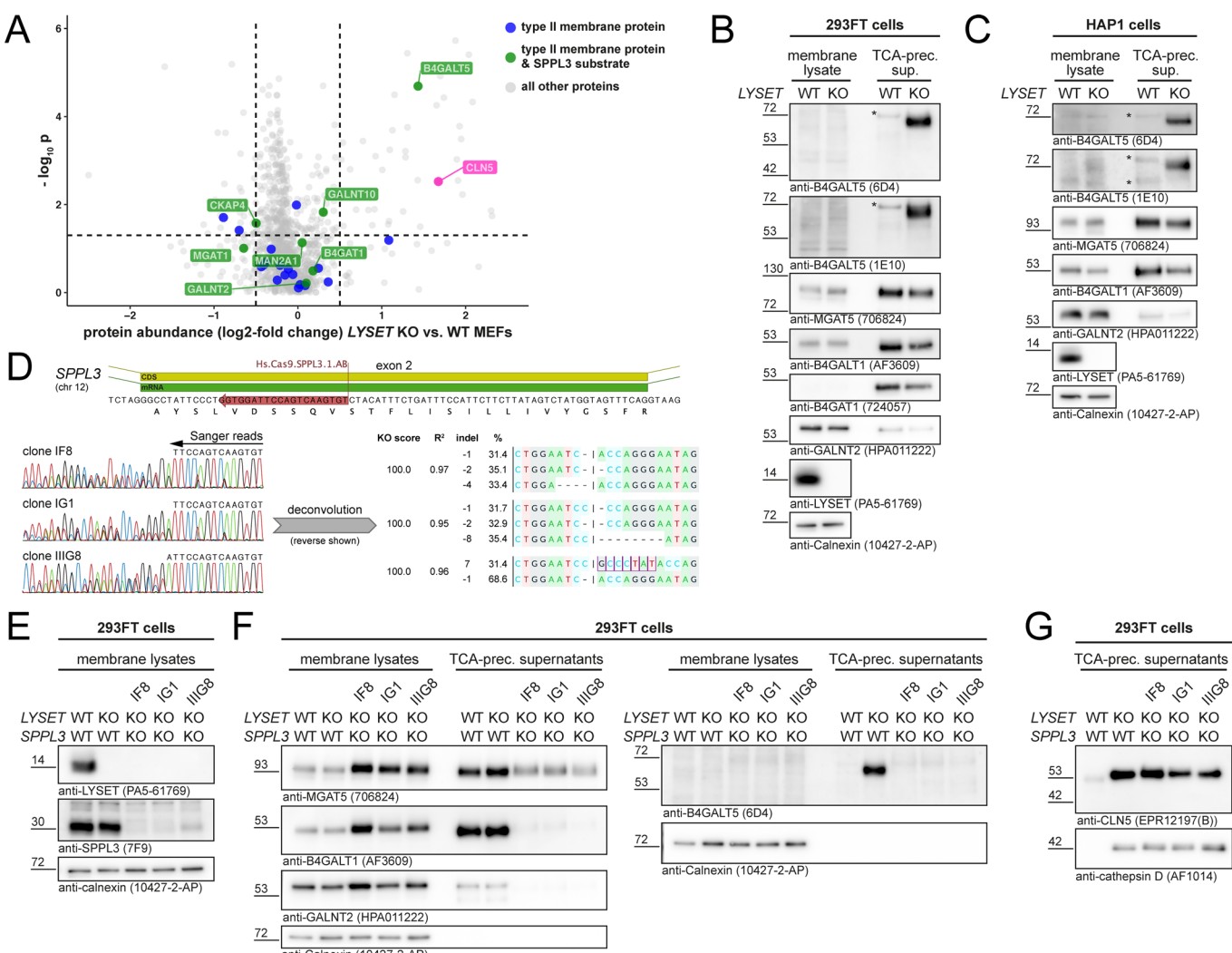

**Figure 1. Loss of LYSET leads to augmented SPPL3-dependent B4GALT5 secretion.**

(A) Re-analysis of published secretome data obtained from *Lyset* KO and WT MEFs (three technical replicates) (Richards et al, 2022). Quantified protein data (unpaired *t* test) from the original study were re-plotted and filtered for type II membrane proteins (blue) and known SPPL3 substrates (green). CLN5 (magenta) is wrongly annotated as type II membrane protein (see G). For consistency, human gene nomenclature is used for murine proteins. Topology annotations were retrieved from Uniprot and SPPL3 substrates were defined as detailed in (Hobohm et al, 2022). (B, C) Glycosyltransferase immunoblots of membrane lysates and conditioned media (supernatants) of parental (WT) and *LYSET* KO 293FT (B) and HAP1 (C) cells. The panels are representative of two and three independent experiments, respectively. Asterisks denote unspecific bands detected with the B4GALT5 antibodies. (D) *SPPL3* targeting strategy implemented in LYSET-deficient 293FT cells using Cas9 genome editing. A crRNA binding the indicated region in *SPPL3* exon 2 was used to knock-out SPPL3 (top center). Editing was confirmed by Sanger sequencing of PCR amplicons of exon 2 in three clones (IF8, IG1 and IIIG8) (bottom left). DECODR (Bloh et al, 2021) was used to pinpoint indels in individual clones (bottom right). (E) Immunoblots confirming the absence of SPPL3 and LYSET from membrane fractions of *LYSET* KO and *LYSET/SPPL3* KO 293FT cells. (F, G) Immunoblot analysis of intramembrane levels and secretion of select glycosyltransferases (F) and lysosomal enzymes (G) from *LYSET*- and *LYSET/SPPL3*-deficient 293FT cells. For (E–G), a single experiment is shown with three biologically distinct *LYSET/SPPL3* KO clones. For all immunoblot panels, calnexin was used as a loading control for membrane fractions. Source data are available online for this figure.

fragment unambiguously demonstrated M6P-tagging of B4GALT5 (Fig. 2F). In contrast, V5-tagged B4GALT1 could not be precipitated using the M6P-binding antibody fragment. As we did not detect intracellular endogenous B4GALT5 by immunoblot, possibly due to low intracellular protein levels or low sensitivity of the antibodies used, we employed the RUSH system (Boncompain et al, 2012) to visualize intracellular maturation of ER-retained B4GALT5-mCherry upon addition of biotin. Recapitulating our observations for secretion of endogenous B4GALT5, *LYSET* KO cells over time secreted B4GALT5-

mCherry following the biotin-induced synchronized release of the reporter, while only a faint release could be detected in media of WT cells (Fig. 2G). Immunoblotting using an mCherry-specific antibody enabled detection of the B4GALT5 reporter in cell lysates and revealed a differential intracellular maturation of B4GALT5 in WT and *LYSET* KO cells following release from the ER, again reflecting the complex glycosylation of B4GALT5 lacking M6P-tags in *LYSET* KO cells. In conclusion, our data reveal that B4GALT5 is subject to M6P-tagging and is targeted to lysosomes for degradation once liberated

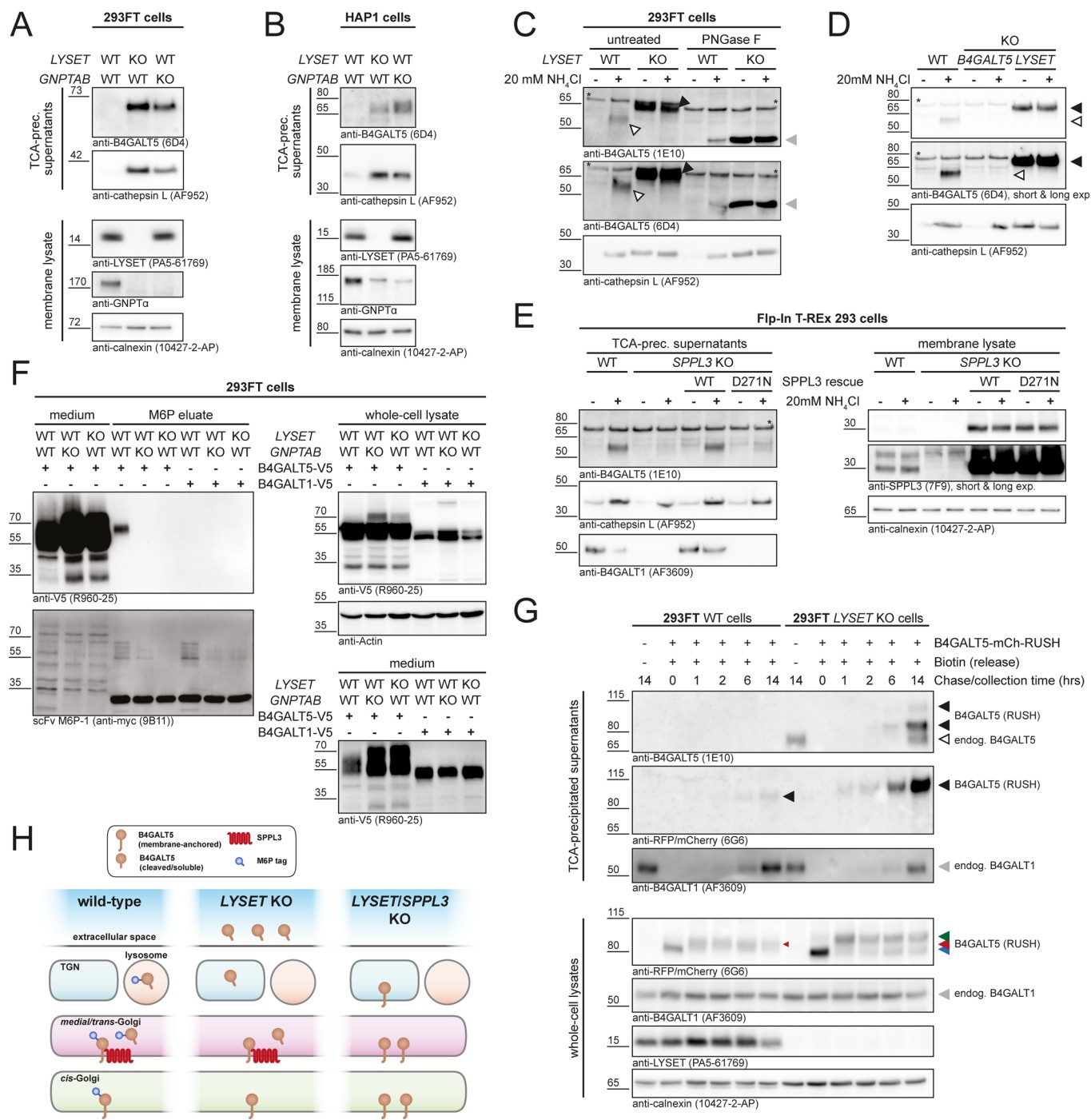

from its membrane anchor by SPPL3-mediated intramembrane proteolysis. These findings explain the SPPL3-dependent B4GALT5 hypersecretion observed in *LYSET* and *GNPTAB* KO cells (Fig. 2H).

## GOLPH3/GOLPH3L-deficiency leads to LYSET destablization, impaired M6P-tagging, and hypersecretion of lysosomal enzymes

The COPI adaptor proteins GOLPH3 and GOLPH3L were recently reported to maintain the *cis-trans* Golgi polarity and proper Golgi

localization of numerous Golgi enzymes, leading to depletion of these enzymes, including B4GALT5, from the Golgi in cells lacking GOLPH3 or both GOLPH3 and GOLPH3L (Rizzo et al, 2021; Welch et al, 2021). To examine whether loss of GOLPH3/ GOLPH3L is also accompanied by hypersecretion of B4GALT5, we established GOLPH3/GOLPH3L-deficient HEK293 cells (Appendix Fig. S3) and performed quantitative comparative whole-cell proteome and secretome analyses of GOLPH3/ GOLPH3L-deficient and parental HEK293 cells (Appendix Fig. S4; Dataset EV1). When specifically filtering our secretome dataset for

◄ **Figure 2. LYSET KO prevents M6P-tagging of B4GALT5 leading to its SPPL3-dependent secretion.**

(A, B) Immunoblots demonstrating B4GALT5 hypersecretion from both *LYSET* and *GNPTAB* KO 293FT (A) (representative of two independent experiments) and HAP1 (B) cells. Conditioned media and membrane fractions were probed with the indicated antibodies. Calnexin was used as loading control for membrane fractions. (C) Ammonium chloride treatment leads to B4GALT5 hypersecretion. Conditioned media of WT and *LYSET* KO 293FT cells incubated overnight with or without 20 mM ammonium chloride were concentrated and subjected to immunoblotting. Enzymatic removal of *N*-glycans using PNGase F was performed where indicated. The electrophoretic mobility of the B4GALT5 proteoform secreted from WT cells (white arrowhead) was distinct from B4GALT5 secreted by *LYSET* KO cells (black arrowhead). This difference was abolished upon *N*-deglycosylation (gray arrowhead). One of two independent experiments is shown, and cathepsin L was used as control. Asterisks denote unspecific bands detected with the B4GALT5 antibodies. (D) The B4GALT5 signal detected in conditioned media from ammonium chloride-treated WT (white arrowhead) and both treated and untreated *LYSET* KO 293FT cells (black arrowhead) was absent from media of ammonium chloride-treated *B4GALT5/B4GALT6* KO 293 cells, demonstrating the specificity of the antibodies used. Asterisks denote unspecific bands detected with the B4GALT5 antibodies. (E) Ammonium chloride-induced B4GALT5 secretion is dependent on SPPL3 activity. Immunoblots of conditioned media and membrane lysates of ammonium chloride-treated parental and *SPPL3* KO Flp-In T-REx 293 cells as well as *SPPL3* KO clones with doxycycline (dox) treatment (100 ng/ml for 48 h)-induced re-expression of SPPL3 WT and the active site mutant D271N. (F) M6P-affinity purification of ectopically expressed B4GALT5 in 293FT cells. WT, *LYSET* KO, or *GNPTAB* KO 293FT cells transfected with B4GALT5-V5 or B4GALT1-V5 expression constructs as indicated were treated with 10 mM ammonium chloride. Media, whole-cell lysates and eluates from anti-M6P immunoprecipitations were analyzed by immunoblot as indicated. (G) Maturation of a B4GALT5-mCherry RUSH reporter in WT and *LYSET* KO 293FT cells. Reporter release was induced by the treatment of cells with 50 µM D-biotin. Samples were collected at indicated time points and subjected to immunoblotting. Endogenous B4GALT1 (gray filled arrowheads), which is not M6P-tagged, was used as control. Filled black arrowheads denote the ectopically expressed B4GALT5 RUSH reporter and the open black arrowhead endogenous B4GALT5. Colored arrowheads illustrate the differential intracellular maturation of the B4GALT5 RUSH reporter: Blue arrowhead, not yet released, i.e., ER-resident B4GALT5; red arrowhead, B4GALT5 released from the ER in 293FT WT cells; green arrowhead, B4GALT5 released from the ER in *LYSET* KO cells carrying more extensive glycosylation. (H) Model explaining B4GALT5 secretion in *LYSET* KO cells. In WT cells, B4GALT5 is subject to M6P-tagging. SPPL3-mediated processing of transmembrane B4GALT5 into a soluble M6P-tagged protein enables its targeting to the endolysosome for degradation (left). Due to impaired M6P-tagging in *LYSET* KO cells, the SPPL3 cleavage product of B4GALT5 is not lysosomally targeted and secreted (center), which is abolished in LYSET- and SPPL3-double-deficient cells and may lead to intra-Golgi accumulation of B4GALT5 (right). Source data are available online for this figure.

type II membrane proteins, we indeed found that secretion of soluble forms of type II membrane proteins was profoundly changed in *GOLPH3/GOLPH3L* KO HEK293 cells (Fig. 3A). Specifically, we also observed a robust hypersecretion of B4GALT5 from *GOLPH3/GOLPH3L* KO HEK293 cells. However, direct comparison revealed that *LYSET* KO cells displayed a substantially stronger increase in B4GALT5 secretion than *GOLPH3/GOLPH3L* KO cells (Fig. 3B). Unexpectedly, we observed a marked loss of LYSET from all four *GOLPH3/GOLPH3L*-deficient HEK293 clones analyzed (Fig. 3B,C). Apart from GOLPH3/GOLPH3L clients such as GPP130/GOLIM4, which was destabilized as reported previously (Welch et al, 2021), very few other integral or peripheral Golgi membrane proteins were reduced in *GOLPH3/GOLPH3L* KO HEK293 cells (Fig. 3C,D), suggesting that the prominent loss of LYSET may be a direct consequence of the loss of GOLPH3/GOLPH3L and not due to global changes within the Golgi membrane proteome. Notably, in *GOLPH3* single KO HEK293 cells, the LYSET protein was stable (Fig. 3E), and levels of established Golgi type II membrane protein GOLPH3/GOLPH3L clients (Welch et al, 2021) were also not reduced (Appendix Fig. S3D). Similar results were obtained in *GOLPH3* KO HAP1 cells (Fig. 3F; Appendix Fig. S5). In HEK293 cells, targeting of the *GOLPH3L* locus alone did also not cause a notable reduction of LYSET and Golgi type II membrane proteins (Appendix Fig. S3E–G), suggesting that GOLPH3 and GOLPH3L have redundant functions. Confocal imaging confirmed that LYSET and GNPT were largely lost from *GOLPH3/GOLPH3L*-deficient HAP1 cells (Fig. 3G,H). Interestingly, we also noted dispersed GM130-immunolabeled *cis*-Golgi compartments in GOLPH3/GOLPH3L-deficient HAP1 cells (Fig. 3G,H), which was supported by electron microscopy revealing a partial destruction and disappearance of mid-Golgi stacks as well as high numbers of transport vesicles surrounding the Golgi stacks (Appendix Fig. S6).

As expected (Pechincha et al, 2022; Richards et al, 2022), reduced LYSET levels in *GOLPH3/GOLPH3L* KO HEK293 cells were also accompanied by a loss of the α-subunit of the GNPT

complex (Fig. 3C). Therefore, we directly assessed M6P-tagging in *GOLPH3/GOLPH3L*-deficient HEK293 and HAP1 cells. To this end, media of ammonium chloride-treated cells were incubated with a myc-tagged anti-M6P single-chain antibody fragment bound to magnetic beads. Following elution with M6P and immunoblot analysis, several M6P-tagged proteins between 20 and 130 kDa were detectable in media of WT cells but neither in media of *GOLPH3/GOLPH3L*, nor of *GNPTAB* and *LYSET* KO 293FT and HAP1 cells (Fig. 4A). Accordingly, only in WT anti-M6P-precipitates the precursor forms of cathepsin L and CLN5 were found. In line with impaired M6P-tagging, *GOLPH3/GOLPH3L* KO HEK293 cells had markedly reduced intracellular levels of mature cathepsin B, L, C, and hexosaminidase B which was accompanied by hypersecretion of the respective precursor forms (Fig. 4B). *GOLPH3/GOLPH3L*-deficient cells similarly released increased amounts of immature precursor forms of CLN5 and cathepsin D and the maturation of CLN5 and the proteolytic processing of the remaining intermediate form to the mature form of cathepsin D was delayed (Appendix Fig. S7A). The impact on soluble lysosomal proteins was also confirmed by our proteome and secretome data (Fig. 4C,D; Appendix Fig. S7B; Dataset EV1). Peptide data further confirmed the immature nature of cathepsins secreted by GOLPH3/GOLPH3L-deficient HEK293 cells (Appendix Fig. S7C; Dataset EV1) and lysosomal enzymes were similarly affected in GOLPH3/GOLPH3L-deficient HAP1 cells (Fig. 4E; Appendix Fig. S7A). Of note, these observations did not apply to β-glucocerebrosidase (GBA, Fig. 4C,D), which is transported to lysosomes in an M6P-independent manner by the lysosomal membrane protein LIMP2 (Reczek et al, 2007). In the absence of M6P-tagging, M6P-independent transport routes via alternative receptors have been demonstrated to assure lysosomal targeting of a subset of lysosomal hydrolases, among them cathepsin D, which might explain the residual mature cathepsin D form in *GOLPH3/GOLPH3L* KO cells (Rijnboutt et al, 1991; Markmann et al, 2015; Braulke et al, 2024). As *GOLPH3* can be amplified in tumors (Scott et al, 2009), we further generated *GOLPH3*- and *GOLPH3/*

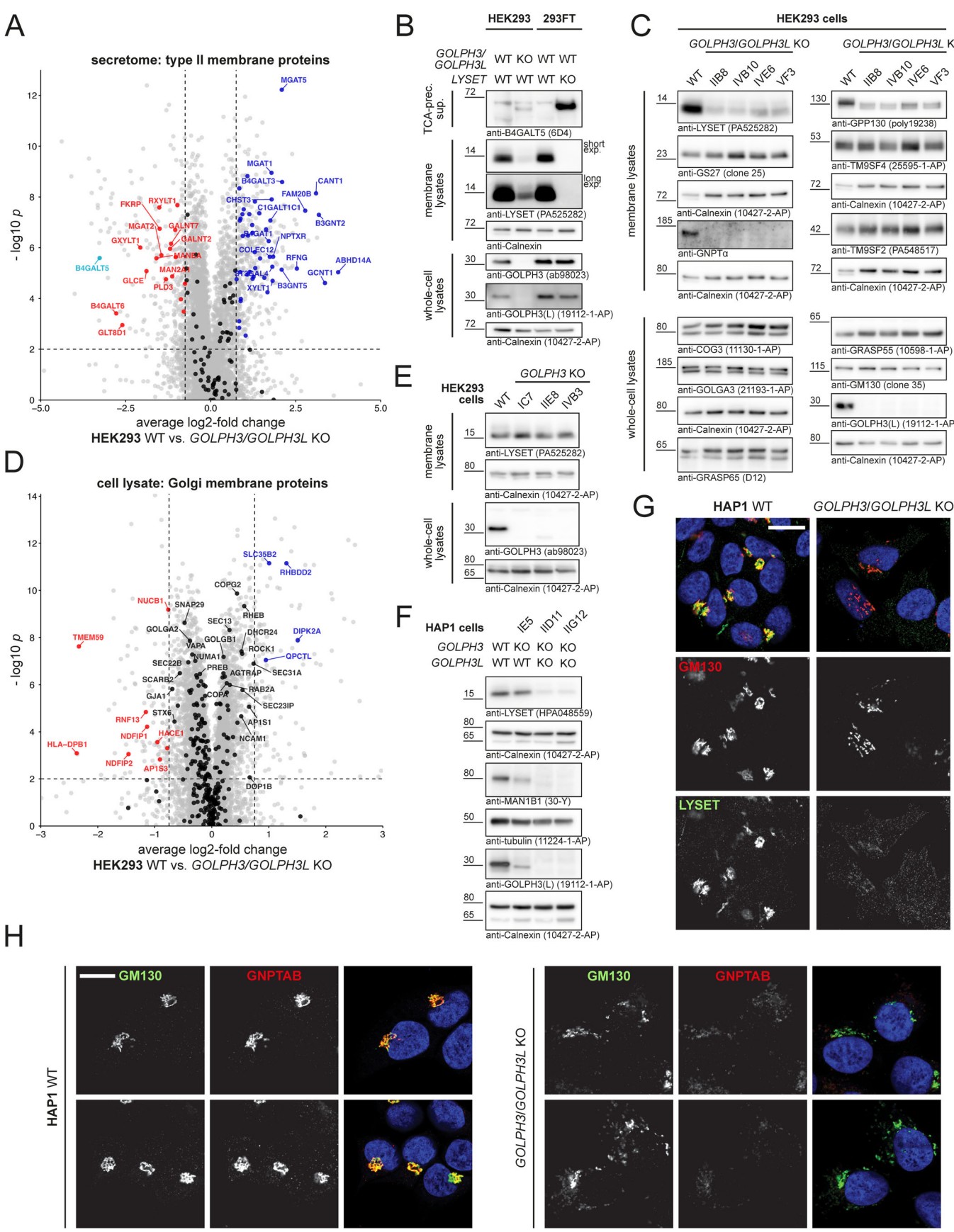

**Figure 3.  Loss of GOLPH3/GOLPH3L leads to reduction of LYSET and GNPT levels.**

(A) Volcano plot depicting mass spectrometric quantification of proteins identified in conditioned media of parental HEK293 cells compared to *GOLPH3/GOLPH3L* KO cells (clone VF3) (*n* = 4 (four individually frozen passages as replicates)). Differential abundance testing (unpaired *t* test) was performed. The full dataset (Appendix Fig. S4B; Dataset EV1) was filtered for type II membrane proteins based on Uniprot annotations. Proteins with significantly (*P* < 0.01) reduced (log2-fold change < −0.75) or increased (log2-fold change >0.75) abundance in parental HEK293 cells are highlighted in red and blue, respectively, and labeled using gene names. B4GALT5 is highlighted in cyan. (B) Comparative analysis of B4GALT5 secretion from *GOLPH3/GOLPH3L*-deficient (clone IIB8), LYSET-deficient, and the corresponding parental cells. TCA-precipitated conditioned supernatants as well as membrane and whole-cell lysates were analyzed by immunoblot. Calnexin was used as a loading control. (C) Immunoblot analysis of Golgi protein abundance in membrane fractions (top panels) and whole-cell lysates (bottom panels) obtained from four biologically distinct *GOLPH3/GOLPH3L* KO clones and parental HEK293 cells. One of three independent experiments is shown. Calnexin was used as a loading control. (D) Volcano plot depicting mass spectrometric quantification of Golgi membrane proteins (excluding type II membrane proteins) identified in whole-cell lysates of parental HEK293 cells compared to *GOLPH3/GOLPH3L* KO cells (clone VF3) (*n* = 4 (four individually frozen passages as replicates)). Differential abundance testing (unpaired *t* test) was performed. The full dataset (Appendix Fig. S4B; Dataset EV1) was filtered for proteins with the GO term annotation "Golgi membrane", but proteins annotated in Uniprot as type II membrane protein were excluded. Significantly changed proteins are colored and labeled as detailed for (A). Non-type II Golgi membrane proteins not significantly changed are shown as black data points, with select points labeled by gene name. (E) LYSET protein levels in membrane lysates of three distinct *GOLPH3* KO clones and parental HEK293 cells. One of two independent experiments is shown. Calnexin was used as loading control. (F) Immunoblotting of whole-cell lysates obtained from HAP1 cells edited in *GOLPH3* and *GOLPH3L*. GOLPH3 and GOLPH3L protein levels in clone IE5 edited in *GOLPH3* as well as clones IID11 and IIG12 edited in *GOLPH3* and *GOLPH3L* (Appendix Fig. S5) were assessed using a GOLPH3 antibody with cross-reactivity towards GOLPH3L. MAN1B1 was previously found to be destabilized in *GOLPH3/GOLPH3L*-deficient cells (Welch et al, 2021). One of three independent experiments is shown. Calnexin served as loading control. (G, H) Confocal Airyscan imaging of WT and *GOLPH3/GOLPH3L* KO HAP1 cells stained with anti-LYSET/anti-GM130 (G) and anti-LYSET/anti-GNPTAB (H). DNA staining (Hoechst 33342) is shown in blue; scale bar = 10 μm. Source data are available online for this figure.

*GOLPH3L*-deficient clones of the non-cancerous diploid human cell line hTERT RPE-1 (Appendix Fig. S8). Again, *GOLPH3/GOLPH3L* KO, but not *GOLPH3* KO, led to a reduction of cellular LYSET levels, hypersecretion of immature lysosomal enzymes, and reduced intracellular abundance of the respective mature proteoforms. The decreased intracellular abundance of soluble lysosomal proteins was also apparent when we re-analyzed proteome data published for *GOLPH3/GOLPH3L* KO U2-OS cells (Welch et al, 2021) (Appendix Fig. S7D). In conclusion, these data demonstrate that *GOLPH3/GOLPH3L*-deficiency in multiple human cell lines is accompanied by a substantial loss of LYSET and GNPT, leading to impaired M6P tagging of soluble lysosomal proteins and ultimately to their aberrant targeting and maturation.

## GOLPH3 maintains Golgi localization of LYSET

Transient (Fig. 5A) or stable, moderate-level cumate-inducible (Fig. 5B) re-expression of WT GOLPH3 in *GOLPH3/GOLPH3L*-deficient HEK293 cells led to restoration of cellular levels of the previously reported GOLPH3/GOLPH3L clients GALNT7 and MAN1B1, but also of LYSET (Fig. 5A,B). In addition, the restoration of mature hexosaminidase B (Fig. 5B) demonstrates a rescue of LYSET function by re-expression of GOLPH3. GOLPH3 was shown to bind Golgi membranes through its phosphatidylinositol-4-phosphate (PI4P)-binding properties located in a positively charged region containing R90 (Dippold et al, 2009; Wood et al, 2009) (Fig. 5C). Moreover, GOLPH3 associates with the coatomer through an N-terminal cluster of arginine residues (R14/R15) (Tu et al, 2012). To examine whether these functional motifs are required for its ability to stabilize LYSET, we similarly re-introduced the GOLPH3 R90L and R14A/R15A mutants into *GOLPH3/GOLPH3L* KO HEK293 cells. GOLPH3 R90L failed to restore cellular MAN1B1, LYSET and mature hexosaminidase B levels when compared to GOLPH3 WT re-expression (Fig. 5B). In contrast, the ability of GOLPH3 R14A/R15A to rescue MAN1B1, LYSET, and hexosaminidase B levels were only slightly reduced compared to GOLPH3 WT. Collectively,

these results indicate that PI4P- and COPI-binding properties of GOLPH3 are required for LYSET stabilization.

In GOLPH3/GOLPH3L-deficient HEK293 cells LYSET accumulated following overnight treatment with Bafilomycin A1, chloroquine or ammonium chloride, which all cause an increase of the lysosomal pH (Fig. 5D). Enrichment of lysosomal and Golgi compartments by differential centrifugation of HEK293 (Fig. 5E) and HAP1 cells (Fig. 5F) revealed that LYSET, but also GNPTα levels were strongly reduced in both the Golgi and the lysosomal fraction of *GOLPH3/GOLPH3L* KO cells. However, following lysosomal protease inhibitor treatment, LYSET was detectable in the lysosomal fraction (Fig. 5E,F). The GNPT α-subunit also accumulated in the lysosomal fraction of inhibitor-treated HAP1 cells, while in HEK293 cells no significant accumulation was noted, possibly owing to lower expression levels, low antibody sensitivity and/or residual activity of different lysosomal proteases (Pechincha et al, 2022). Upon high-resolution immunofluorescence imaging of parental HAP1 cells, LYSET was almost exclusively found in the Golgi apparatus (Fig. 3G) and consequently did not colocalize with the lysosomal membrane protein LAMP2 (Fig. 5G; Appendix Fig. S9A). In contrast, LYSET-specific staining was largely lost from *GOLPH3/GOLPH3L* KO HAP1 cells, with only little LYSET remaining in GM130-positive regions (Figs. 3G and 5G; Appendix Fig. S9A). Notably, when *GOLPH3/GOLPH3L* KO HAP1 cells were treated with protease inhibitors, a brighter, vesicular LYSET staining was observed, and LYSET was partially co-localizing with LAMP2-positive lysosomes (Fig. 5G; Appendix Fig. S9A). Reminiscent of the observations we reported for LYSET-deficient cells (Richards et al, 2022), also the GNPT complex disappeared from *GOLPH3/GOLPH3L* KO cells and was mostly found in lysosomes upon treatment with protease inhibitors (Fig. 5G,H; Appendix Fig. S9). Taken together, this demonstrates that Golgi localization of LYSET and, as a consequence, the GNPT complex, is abolished in the absence of GOLPH3/GOLPH3L and LYSET is subsequently trafficked to and degraded in lysosomes. Degradation of LYSET in lysosomes is inhibited by increased lysosomal pH or pre-treatment with lysosomal protease inhibitors.

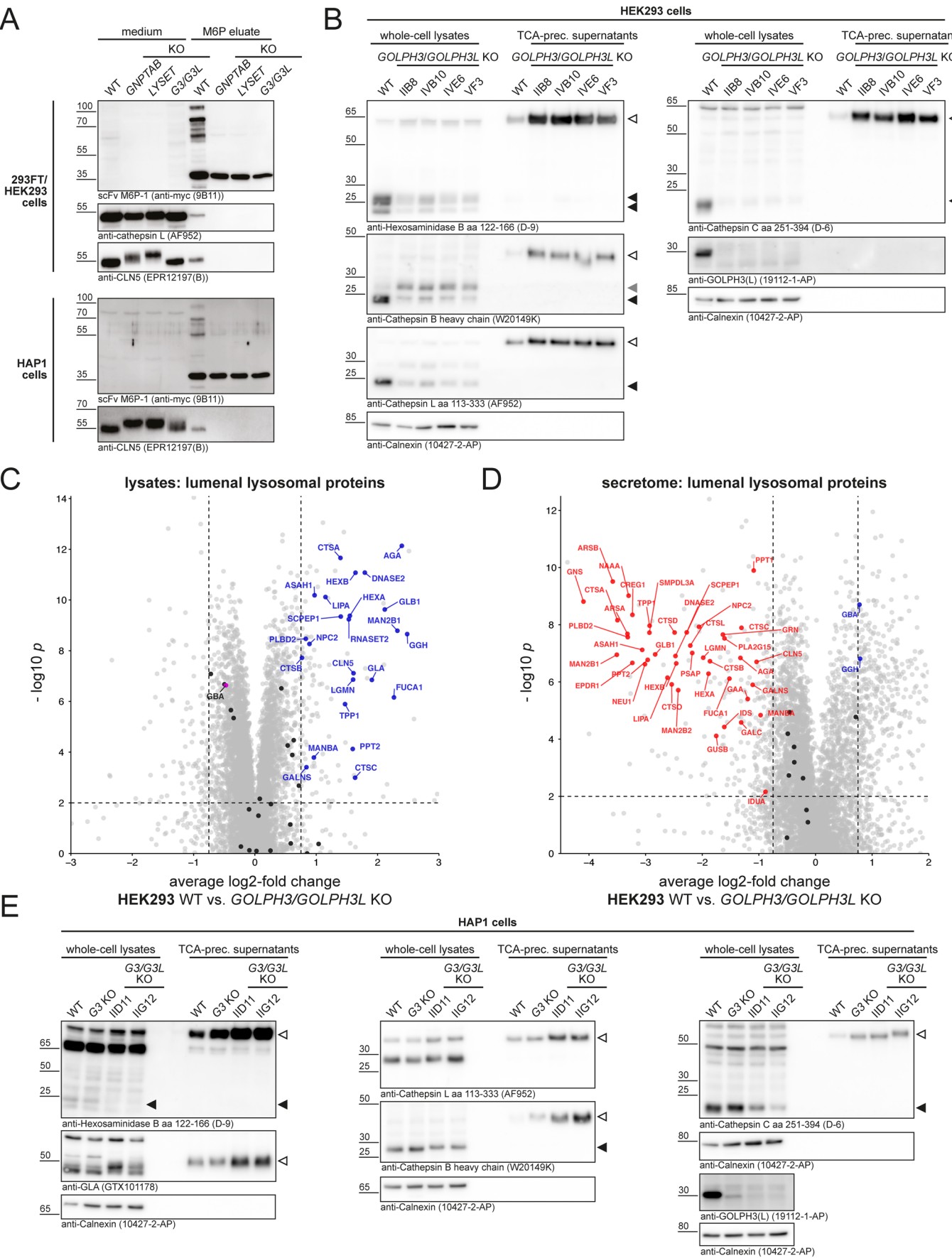

**Figure 4. Loss of LYSET in *GOLPH3/GOLPH3L*-deficient cells results in defective M6P-tagging and leads to hypersecretion of soluble lysosomal enzymes.**

(A) Impaired M6P labeling in *GOLPH3/GOLPH3L* KO HEK293 cells (top panels) and HAP1 cells (bottom panels). Tenfold concentrated conditioned media from the indicated cell lines treated with 10 mM ammonium chloride for 24 h were subjected to anti-M6P immunoprecipitation. Fractions eluted with M6P were analyzed by immunoblotting for total M6P-labeled proteins and the control proteins cathepsin L and CLN5. (B) Immunoblot analysis of intracellular abundance and secretion of lysosomal enzymes in *GOLPH3/GOLPH3L*-deficient HEK293 clones and WT control cells. Whole-cell lysates and TCA-precipitated supernatants were stained with antibodies against the indicated lysosomal enzymes. Mature proteoforms: filled black arrowheads, incompletely processed proteoforms: filled gray arrowhead, immature secreted precursor forms: open arrowheads. One of two independent experiments done with four biologically distinct *GOLPH3/GOLPH3L* KO clones is shown. Calnexin served as a loading control (see also Appendix Fig. S7A). (C, D) Cellular proteome (C) and secretome (D) datasets (Appendix Fig. S4; Dataset EV1) filtered for lysosomal proteins ($n = 4$ (four individually frozen passages as replicates)). Differential abundance testing (unpaired *t* test) was performed. Select well-characterized luminal lysosomal proteins (Markmann et al, 2017) (Dataset EV1) are highlighted. Data points for significantly ($P < 0.01$) reduced (log2-fold change $< -0.75$) or increased (log2-fold change $>0.75$) soluble lysosomal proteins are shown in red and blue, respectively, and labeled by gene name. Soluble lysosomal proteins not fulfilling these criteria are shown in black and all other quantified proteins in gray in the background. β-glucocerebrosidase (GBA) is highlighted in purple in (C). (E) Immunoblot analyses of select lysosomal enzymes in lysates and conditioned media of parental, *GOLPH3* KO, and *GOLPH3/GOLPH3L* KO HAP1 cells. Open arrowheads denote immature precursor forms of the lysosomal enzymes, and filled black arrowheads denote the mature proteoform of the lysosomal enzymes. One of two independent experiments done with four biologically distinct *GOLPH3/GOLPH3L* KO clones is shown. Calnexin served as a loading control for the membrane fractions (see also Appendix Fig. S7A). Source data are available online for this figure.

## LYSET is a novel atypical GOLPH3/GOLPH3L client

Molecularly, the peripheral Golgi membrane proteins GOLPH3 and GOLPH3L are believed to facilitate retrieval of clients through interactions with the COPI coat on the one hand and their clients' cytosolic tails on the other hand (Schmitz et al, 2008; Tu et al, 2008; Chang et al, 2013; Isaji et al, 2014; Pereira et al, 2014; Eckert et al, 2014; Welch et al, 2021; Rizzo et al, 2021). We could not directly assess co-localization of cell-endogenous GOLPH3/GOLPH3L and LYSET due to a lack of sensitive antibodies from different host species, but endogenous GNPT and GOLPH3 co-localized in HAP1 cells (Fig. 5H; Appendix Fig. S9B). All known GOLPH3/GOLPH3L clients are type II membrane proteins which harbor positively charged amino acids in their short cytoplasmic, N-terminal regions directly preceding their transmembrane domain (TMD) (Rizzo et al, 2021; Welch et al, 2021). LYSET deviates markedly from known clients, as it is annotated as double-pass membrane protein. However, its cytosolic N-terminus harbors numerous positively charged amino acid residues (Fig. EV1A). We therefore used Alphafold2 to model a potential interaction of GOLPH3 and LYSET (Fig. EV1B,C). With its PI4P-binding pocket (formed by e.g. R90 and R171) facing the cytosolic membrane boundary, GOLPH3 was predicted to primarily be in contact with the arginine-rich cytoplasmic region of LYSET immediately preceding its TMD1 and with the loop region directly following LYSET TMD2 (Fig. EV1C,D). In this model, LYSET R29, R32, R37, and R39 are in close contact with a negative surface region formed by GOLPH3 D247, E250, D258, and D262 (Fig. EV1C,D) which had previously been suspected as client binding site (Welch et al, 2021). Taken together, these structural predictions further suggested that membrane-proximal positively charged amino acid residues in the LYSET N-terminal cytosolic region could facilitate GOLPH3 binding.

To test whether GOLPH3 is able to bind LYSET, we performed co-immunoprecipitation experiments. Indeed, LYSET was co-precipitated from lysates of cells transfected with both GOLPH3 and LYSET following GOLPH3 immunoprecipitation with two distinct commercial antibodies (Figs. 6A and EV1E). We additionally observed co-precipitation of endogenous GOLPH3 with the LYSET–GNPT-complex in HEK293 cells overexpressing both LYSET-FLAG and GNPTAB-myc in a doxycycline-inducible manner (Fig. 6B). Overexpressed GOLPH3 also co-precipitated

cell-endogenous COPB (Fig. 6A), a subunit of the COPI coatomer and a known interactor of GOLPH3 (Tu et al, 2012; Welch et al, 2021). Interestingly, whereas LYSET-FLAG was strongly enriched compared to input levels in the FLAG-immunoprecipitation, this was not the case for GOLPH3, which is in line with GOLPH3 having multiple competing Golgi membrane proteins clients (Welch et al, 2021) besides LYSET. In conclusion, our data support that LYSET is a novel, non-type II membrane protein client of GOLPH3/GOLPH3L.

Next, we transiently expressed LYSET constructs, in which individual arginine residues in the N-terminal cytosolic region predicted to be in close contact with GOLPH3/GOLPH3L had been mutated, in HEK293 cells. Replacement of the membrane-distal residues R22, R29 or R32 by alanine and also the R39W (known to cause a mucolipidosis type II/III-like phenotype in humans (Ain et al, 2021)) and R39A mutations had no impact on LYSET protein levels following transient transfection (Figs. EV1F and 6C). However, in case of the R37A mutant, LYSET levels in cell lysates of transfectants were markedly reduced compared with LYSET WT and this was similarly observed for the R22A/R37A, R29A/R37A and R37A/R39A double mutants and to a lesser extent for the R32A/R37A double mutant (Fig. EV1F). A double mutant, in which arginine residues R37 and R39 were replaced by positively charged lysines displayed no reduced expression in transfected cells (Fig. 6C). Using CRISPR/Cas9 genome editing, we then established LYSET R37A/R39A knock-in hTERT RPE-1 and HEK293 cells (Fig. 6D). In agreement with the overexpression data, LYSET levels in R37A/R39A knock-in hTERT RPE-1 (Fig. 6E) and HEK293 (Fig. 6F) clones were markedly reduced compared to the parental cells. Moreover, knock-in cells hypersecreted immature hexosaminidase B and cathepsin L and displayed markedly reduced intracellular levels of the mature proteoforms of these enzymes, although this effect was not as pronounced as in *LYSET* KO clones. Collectively, our transfection and knock-in data show that R37, one of the arginine residues in the N-terminal cytosolic region predicted to facilitate an interaction with GOLPH3/GOLPH3L, is a key determinant of LYSET stability. Whether this is a direct consequence of reduced interaction with GOLPH3/GOLPH3L remains to be elucidated.

Finally, considering that LYSET interacts with and stabilizes GNPTAB (Richards et al, 2022; Pechincha et al, 2022) we used the more recent Alphafold 3 (Abramson et al, 2024) to predict a

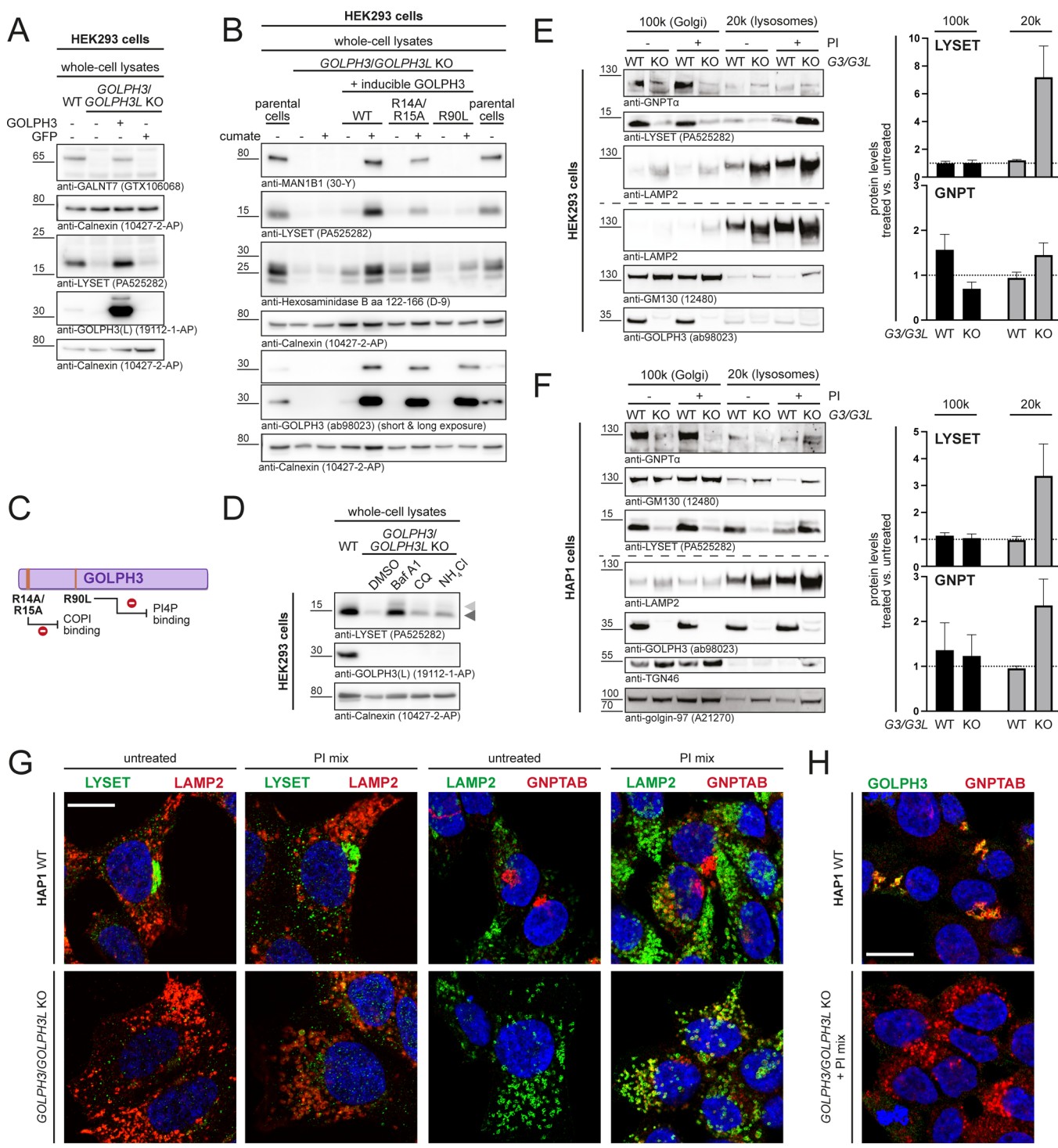

putative ternary complex formed by GOLPH3, LYSET and GNPTAB (Figs. 6G–I and EV2). In agreement with the Alphafold2 prediction, the arginine-harboring N-termini of LYSET isoform 1 but also isoform 2 were predicted to be in contact with the negatively charged surface of GOLPH3. Interestingly, however, Alphafold 3 also predicted that the TMD close to the C-terminus of the β-subunit of GNPT is in proximity to TMD1 and TMD2 of LYSET and that positively charged arginine residues in the

C-terminal cytosolic tail of the GNPT β-subunit are positioned above a separate negatively charged patch on the GOLPH3 surface (Figs. 6G–I and EV2C–E). Additional modeling suggested that dimerization of GNPTAB would not interfere with the formation of a GOLPH3:LYSET:GNPT complex, as the dimer interface, in agreement with recent cryo-EM data of a soluble GNPT construct (Li et al, 2022), was primarily formed by ectodomain contacts (Fig. EV2F). However, the predicted interaction does not take into

**Figure 5. LYSET is mislocalized and lysosomally degraded in GOLPH3/GOLPH3L-deficient cells.**

(A) Transient transfection of GOLPH3 WT into *GOLPH3/GOLPH3L* KO HEK293 cells (clone VF3) restores cellular GALNT7 and LYSET levels in whole-cell lysates. One of three independent experiments is shown. Calnexin was used as a loading control. (B) Cumate-induced re-expression of GOLPH3 WT, R14A/R15A, and R90L in stably transfected *GOLPH3/GOLPH3L* KO HEK293 cells (clone VF3). Whole-cell lysates were analyzed by immunoblotting. One of two independent experiments is shown. Calnexin was used as a loading control. (C) Schematic overview of GOLPH3 and the reported impact of the GOLPH3 point mutations used. (D) Overnight treatment of WT and *GOLPH3/GOLPH3L* KO HEK293 cells (clone VF3) with the lysosomal inhibitors bafilomycin A1 (Baf A1; 200 nM), chloroquine (CQ; 100 μM) or ammonium chloride (NH$_4$Cl; 20 mM). Whole-cell lysates were analyzed by immunoblot. Light gray and dark gray arrowheads denote LYSET isoform 1 (long) and isoform 2 (short), respectively. Calnexin was used as loading control. (E, F) LYSET and GNPTα immunoblot analysis of Golgi- and lysosome-enriched fractions obtained by differential centrifugation at 100,000×*g* (100k) and 20,000×*g* (20k), respectively, of HEK293 (E) and HAP1 (F) cells. Where indicated cells were pre-treated for 24 h with protease inhibitors (PI; 5 μg/mL E64d, 100 μM Leupeptin, 100 μM Pepstatin A). GM130, golgin97 and TGN46 served as Golgi markers, LAMP2 as lysosomal marker. Different membranes blotted with the same samples are indicated by dashed lines. Right: Immunoblot quantification (n = 3 (independent experiments)) of changes in LYSET and GNPTAB levels in Golgi and lysosomal fractions upon protease inhibitor treatment. Data represent mean values + SEM. (G) Immunofluorescence staining and AiryScan imaging of WT and *GOLPH3/GOLPH3L* KO (clone IID11) HAP1 cells. Cells were treated with protease inhibitors (PI mix) overnight where indicated. Scale bar = 10 μm. (H) GOLPH3 and GNPTAB immunofluorescence staining in HAP1 cells demonstrating co-localization in the Golgi apparatus in WT cells. GOLPH3 staining is absent in *GOLPH3/GOLPH3L* KO cells, in which GNPTAB staining is detected in lysosomes upon PI treatment. Scale bar = 10 μm. Single-channel images of (G, H) are compiled in Appendix Fig. S9. Source data are available online for this figure.

account the activation of GNPTAB by site-1-protease-mediated cleavage (Marschner et al, 2011). Interestingly, in this predicted complex, the GNPTα N-termini, which include a stretch of positively charged amino acids previously reported to directly interact with the COPI coat (Liu et al, 2018), are similarly found in close proximity of the GOLPH3 adapter proteins. In sum, these predictions raise the possibility that the COPI adapters GOLPH3 and GOLPH3L may physically interact with LYSET but also with GNPT subunits.

## Discussion

Along with Golgi cargo, glycosyltransferases, and other Golgi enzymes are subject to continuous anterograde trafficking. Therefore, selective COPI vesicle-mediated retrograde transport of Golgi enzymes is required to maintain their precise Golgi subcompartment-specific localization at steady state. Acting as COPI adaptors to facilitate retrograde transport, GOLPH3 and GOLPH3L interact with the cytoplasmic tails of select Golgi enzymes to maintain their Golgi localization (Schmitz et al, 2008; Tu et al, 2008; Chang et al, 2013; Isaji et al, 2014; Pereira et al, 2014; Eckert et al, 2014; Welch et al, 2021; Rizzo et al, 2021). Rizzo and colleagues for instance reported GOLPH3-dependent Golgi retention of B4GALT5, a mid-Golgi-localized galactosyltransferase acting early in glycosphingolipid biosynthesis (D'Angelo et al, 2013), and lysosomal degradation of B4GALT5 upon *GOLPH3* knockdown (Rizzo et al, 2021). In addition, Welch et al. found that GOLPH3 and its paralogue GOLPH3L physically bound numerous Golgi enzymes, many of which were destabilized in *GOLPH3/ GOLPH3* double-deficient cells (Welch et al, 2021).

Here, we have identified a novel aspect of Golgi enzyme trafficking based on our observations that the soluble ectodomain of B4GALT5 is (i) hypersecreted by both LYSET- and GNPT-deficient cells and (ii) precipitated with an anti-M6P single-chain antibody fragment from media collected from multiple ammonium chloride-treated WT cell lines. These data unambiguously demonstrate that the Golgi glycosyltransferase B4GALT5 is subject to M6P-tagging and can thus be targeted to lysosomes, most likely for degradation. Supporting this further, previous proteomic studies have observed that a soluble form of B4GALT5 was increased in the serum of mice deficient for the cation-independent MPR (Qian

et al, 2008) and enriched by affinity purification of M6P-tagged proteins from differentiated osteoclast culture supernatants and serum (Czupalla et al, 2006; Sleat et al, 2013). In addition to B4GALT5, several other type I and type II membrane proteins have been reported to be M6P-tagged by unbiased proteomic studies (Czupalla et al, 2006; Qian et al, 2008; Sleat et al, 2013; Čaval et al, 2019; Huang et al, 2019; Richards et al, 2022). However, apart from such mass spectrometric observations, the consequences of M6P tagging of these membrane proteins for their targeting and localization have not been investigated. With B4GALT5, our study thus characterizes a first prototypic membrane protein that is tagged with M6P, possibly to prevent its secretion and target it for degradation in lysosomes.

Notably, the membrane anchor of the type II membrane protein B4GALT5 along with its cytosolic N-terminus, which ensures GOLPH3-dependent retrieval of B4GALT5 within the Golgi (Rizzo et al, 2021), would likely preclude M6P-dependent lysosomal sorting. Intramembrane proteolysis catalyzed by SPPL3, which cleaves a wide range of Golgi type II membrane proteins (Voss et al, 2014; Kuhn et al, 2015; Jongsma et al, 2021; Hobohm et al, 2022; Yang et al, 2023), would release the B4GALT5 ectodomain into the Golgi lumen, preventing retrieval and enabling M6P-dependent transport to lysosomes (Fig. 7A). Indeed, further establishing B4GALT5 as an SPPL3 substrate, B4GALT5 accumulated in Golgi-enriched fractions of *SPPL3*-deficient cells (Hobohm et al, 2022) and *B4GALT5* KO could reverse glycosphingolipid-dependent phenotypes instigated by *SPPL3* KO (Jongsma et al, 2021; Yang et al, 2023). Importantly, the molecular determinants which selectively predispose B4GALT5 but not B4GALT1, another SPPL3 substrate, to M6P- tagging are presently unclear. Moreover, it remains to be determined whether M6P-tagging occurs prior to or after cleavage of B4GALT5 by the mid-Golgi-localized SPPL3 (Truberg et al, 2022). Furthermore, it is still unclear to what extent the soluble B4GALT5, like M6P-tagged lysosomal hydrolases, is partially secreted and recaptured by surface MPR (Braulke et al, 2024), since it was previously identified as an SPPL3 substrate in the secretome of LYSET-proficient murine cells (Kuhn et al, 2015). Notably, its M6P-tagging clearly sets B4GALT5 apart from other reported SPPL3 substrates that are readily secreted from WT cells (Voss et al, 2014; Kuhn et al, 2015; Hobohm et al, 2022) and can also be detected in human serum (Hobohm et al, 2022). This obviously raises the question of whether other SPPL3 substrates can

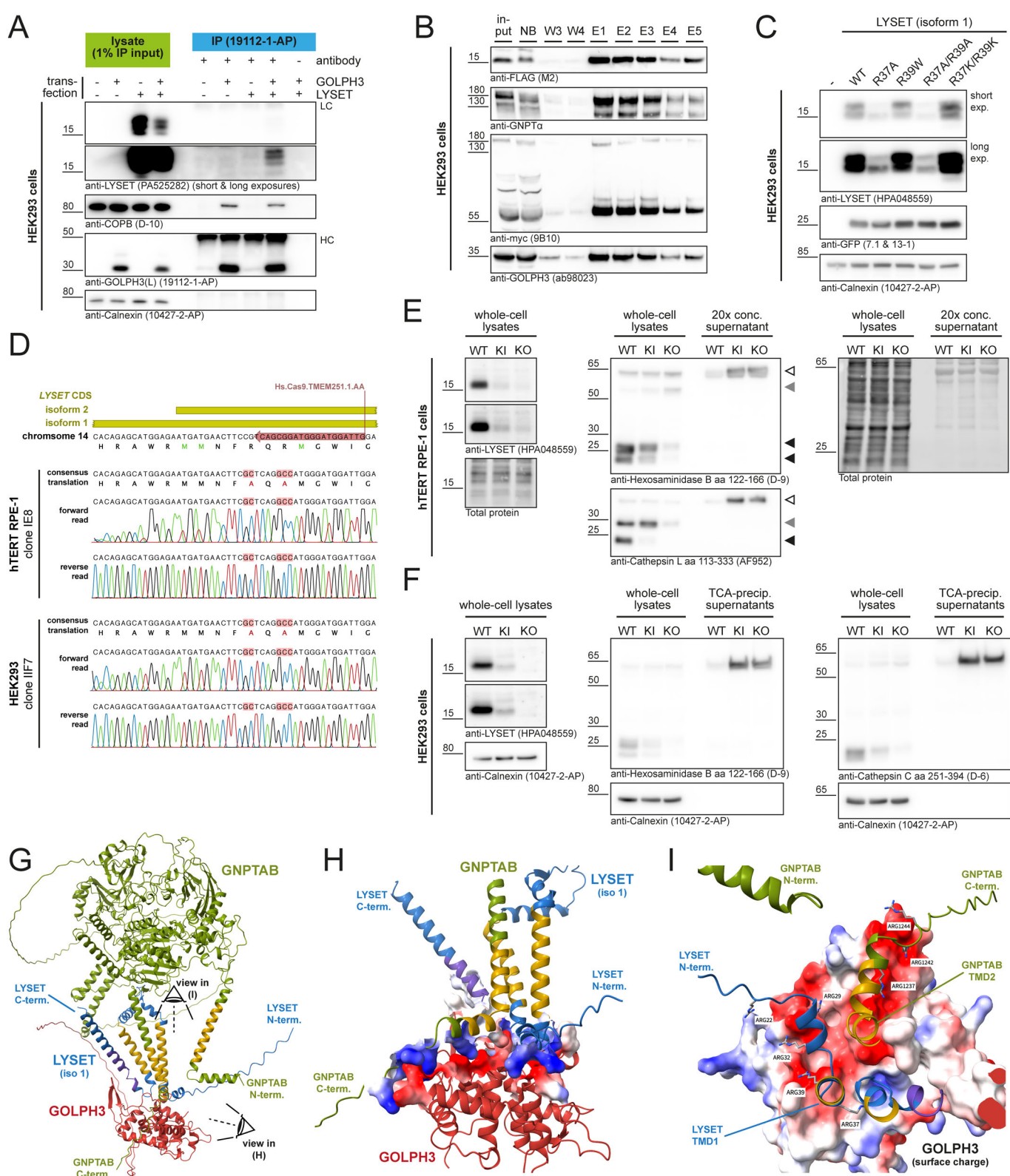

similarly be subject to M6P-tagging. Although it is still unclear whether Golgi enzymes fulfill an important physiological function following their SPPL3-mediated cleavage and secretion, it is intriguing to speculate that B4GALT5 is specifically targeted to

lysosomes by M6P tagging to prevent its release into the extracellular space.

In addition, we demonstrate here that the simultaneous loss of GOLPH3 and GOLPH3L not only affects Golgi glycosyltransferase

Figure 6.   LYSET is an atypical GOLPH3/GOLPH3L client.

(A) Co-immunoprecipitation of LYSET and GOLPH3 in HEK293 transfectants. WT HEK293 cells were transfected as indicated, and GOLPH3 was immunoprecipitated with anti-GOLPH3 (19112-1-AP) from cell lysates. Immunoprecipitates were analyzed by immunoblotting. One of three independent experiments is shown. Calnexin was used as a loading control for lysate input samples. LC antibody light chain, HC antibody heavy chain. (B) Co-immunoprecipitation of LYSET-FLAG and GOLPH3 from HEK293 cells overexpressing LYSET-FLAG and GNPTAB-myc. LYSET-FLAG was bound to FLAG-agarose and eluted using FLAG-peptide. Input (1%), flow-through (NB), wash (W) and elution fractions (E) were analyzed by immunoblotting using indicated antibodies. (C) Transient overexpression of LYSET arginine mutants. HEK293 cells were transfected with plasmids encoding the indicated LYSET variants and LYSET protein levels were analyzed by immunoblotting. A co-transfected GFP plasmid served as transfection control and calnexin as loading control. One of three independent experiments is shown. (D) Sanger sequencing confirmation of the successful knock-in of the LYSET R37A/R39A variant in hTERT RPE-1 (top) and HEK293 (bottom) cells. Forward and reverse reads of the PCR-amplified target region of LYSET were aligned to the human reference genome. The binding site of the crRNA used is indicated. (E, F) LYSET protein levels and lysosomal enzyme maturation/secretion in LYSET R37A/R39A knock-in (KI) hTERT RPE-1 (E) or HEK293 (F) cells. Parental and LYSET KO cells (Appendix Fig. S10) served as controls. Mature proteoforms of lysosomal enzymes: filled black arrowheads, incompletely processed proteoforms: filled gray arrowheads, immature secreted precursor forms: open arrowheads. One of four independent experiments is shown. Total protein staining or calnexin signals served as loading control. (G) AlphaFold 3 prediction of a ternary complex of GOLPH3, LYSET (isoform 1) and GNPTAB. Annotated TMD regions are colored in yellow. In the case of LYSET, a putative TMD region predicted by DeepTMHMM (Hallgren et al, 2022) is shown in purple. (H) Side view of the complex shown in (G), highlighting close proximity of GOLPH3 and the LYSET N-terminus as well as the GNPTAB C-terminus. Surface electrostatic potential is shown for selected amino acid side chains. (I) Top-down view from the membrane plane onto the GOLPH3 surface. Positively charged amino acid side chains in LYSET and GNPTAB are shown as stick representation. Source data are available online for this figure.

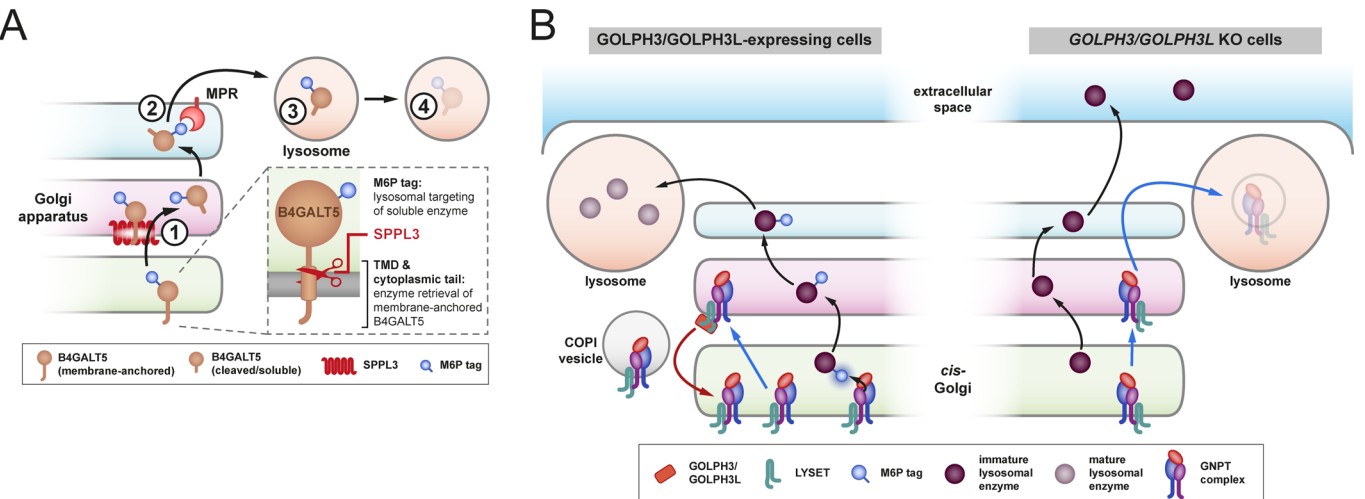

Figure 7.   Schematic summary.

(A) In the Golgi, B4GALT5 is cleaved off its membrane anchor by SPPL3 (1). As it is M6P-tagged, soluble B4GALT5 is recognized by mannose 6-phosphate receptors (MPR) (2) and transported to lysosomes (3), where it likely is degraded (4). (magnification) B4GALT5 carries two competing sorting motifs. Intact, full-length B4GALT5 can be retained in the Golgi due to sorting motifs in its transmembrane domain (TMD) or cytoplasmic N-terminus. SPPL3 cleavage enables lysosomal targeting via M6P. (B) Localization of the GNPT complex and LYSET with implications for M6P-dependent targeting of lysosomal enzymes in WT (left) and GOLPH3/GOLPH3L-deficient (right) cells. In WT cells, the GNPT complex (along with LYSET) is largely found in cis-Golgi cisternae, where it initiates M6P-tagging of lysosomal enzymes which are then transported to lysosomes in an MPR-dependent manner (black arrows). Our data suggest that this steady-state localization is actively maintained by retrieval in COPI vesicles by GOLPH3/GOLPH3L (red arrow). Absence of GOLPH3/GOLPH3L results in transport of GNPT/LYSET to lysosomes where they are degraded (right, blue arrow) causing impaired M6P-tagging and hypersecretion of immature lysosomal enzymes (right, black arrows).

trafficking but also results in a profound destabilization of LYSET and GNPT, causing a general defect in M6P-tagging of both soluble lysosomal enzymes and select integral membrane proteins such as B4GALT5. Complementing our previous work on the crucial role of LYSET for Golgi localization and stabilization of the GNPT complex (Pechincha et al, 2022; Richards et al, 2022), our data reveal a molecular mechanism underlying the Golgi localization of LYSET and explain why a previously reported deletion of GOLPH3 alone did not affect GNPT localization (Liu et al, 2018). As genetic screens generally are focussed on phenotypic changes caused by single-gene KO, this may similarly explain why LYSET but neither GOLPH3 nor GOLPH3L has been found as relevant hit that impaired lysosomal function in recent genome-wide screens

performed by us and others (Pechincha et al, 2022; Richards et al, 2022; Zhang et al, 2022). Indeed, our observations support the notion that GOLPH3 and GOLPH3L are functionally redundant in regard to their role in Golgi enzyme retrieval and that, in the cell culture models tested, endogenous expression of either GOLPH3 or GOLPH3L is necessary but also sufficient to maintain steady-state levels of LYSET but also other previously established clients. The relevance of the direct interaction of the GNPT α-subunit with COPI (Liu et al, 2018) for proper localization and function of GNPT in the Golgi still needs to be explored, since LYSET-mediated retention and stabilization appear to be dominant. Furthermore, structural predictions also place the previously identified COPI-binding region in close proximity to the COPI

adapter GOLPH3. The observation that stable LYSET expression depends on cellular GOLPH3/GOLPH3L levels and function strongly suggests that LYSET and thus GNPT are constantly recycled within the Golgi apparatus and hence rely on GOLPH3/GOLPH3L-dependent active retrieval in COPI vesicles (Fig. 7B). Established GOLPH3/GOLPH3L clients feature cytosolic N-termini rich in positive charges that likely enable molecular interactions with GOLPH3/GOLPH3L (Rizzo et al, 2021; Welch et al, 2021). Indeed, LYSET also has an N-terminal cytosolic region rich in basic amino acids and could thus, despite its different topology, be subject to a similar GOLPH3/GOLPH3L-dependent retrieval mechanism. Interestingly, two distinct isoforms of LYSET have been reported that differ in the length of their cytosolic N-terminus but are equally capable of rescuing a LYSET KO phenotype, suggesting that the short N-terminus of the smaller isoform 2 is sufficient to ensure LYSET stability. Supporting this, our experimental data show that in particular the membrane-proximal R37 is of importance for the maintenance of stable cellular LYSET levels. Finally, while strongly supporting a molecular interaction with LYSET, structural predictions further suggest a simultaneous molecular interaction of GOLPH3/GOLPH3L with the GNPT β-subunit, raising the possibility that COPI-dependent retrieval and thus Golgi localization of the GNPT complex is ensured through the formation of a multimeric complex, which needs to be further dissected in the future.

In conclusion, although M6P-tagging of integral membrane proteins from different subcellular compartments has been previously described (Czupalla et al, 2006; Qian et al, 2008; Sleat et al, 2013; Čaval et al, 2019; Huang et al, 2019; Richards et al, 2022), we now unambiguously demonstrate the presence and cellular function of M6P tagging of the soluble ectodomain of B4GALT5 and its M6P-dependent intracellular retention. Moreover, our study delineates the molecular mechanism ensuring *cis*-Golgi localization of LYSET and is thus of fundamental significance for the understanding of the organization of the M6P-tagging machinery that ensures lysosomal function and integrity.

# Methods

### Reagents and tools table

| Reagent/resource | Reference or source | Identifier or catalog number |
| --- | --- | --- |
| **Experimental models** | | |
| 293FT cells | Thermo Fisher Scientific | R70007 |
| 293FT cells (*LYSET* KO) | Richards et al (2022) | – |
| 293FT cells (*GNPTAB* KO) | Richards et al (2022) | – |
| Flp-In T-REx 293 cells | Thermo Fisher Scientific | R78007 |
| HAP1 cells | Horizon Discovery | C631 |
| HAP1 cells (*LYSET* KO) | Horizon Discovery | HZGHC007846c002 |

| Reagent/resource | Reference or source | Identifier or catalog number |
| --- | --- | --- |
| HAP1 cells (*GNPTAB* KO) | Horizon Discovery | HZGHC001785c004 |
| HEK293 cells | ATCC | CRL-1573 |
| HEK293 (*SPPL3* KO) cells | Hobohm et al (2022) | – |
| HEK293 (*B4GALT5/B4GALT6* KO) | Narimatsu et al (2019) | – |
| HEK293 Tet-On 3G cells | TaKaRa | 631182 |
| hTERT RPE-1 cells | ATCC | CRL-4000 |
| **Recombinant DNA** | | |
| LYSET isoform 1 and isoform 2 plasmids | Richards et al (2022) | – |
| pENTR1A | Thermo Fisher Scientific | A10462 |
| pcDNA5/FRT/TO/DEST-SPPL3 | Hobohm et al (2022) | – |
| pENTR1A-GFP-N2 (FR1) | Addgene | 19364 |
| pBQM812A | System Biosciences | PBQM812A-1 |
| Str-KDEL-flGALT-SBP-tagBFP | Addgene | 65274 |
| Str-KDEL_ST-SBP-mCherry | Addgene | 65265 |
| pcDNA3.1-GNPTAB | Addgene | 78108 |
| pcDNA6/myc-His B | Thermo Fisher Scientific | V22120 |
| pLenti_CMVTRE3G-Puro-DEST | Addgene | 27565 |
| pEF-DEST51-GOLPH3 | Addgene | 21690 |
| pOG44 | Thermo Fisher Scientific | V600520 |
| **Antibodies** | | |
| B4GALT5 6D4 | Steentoft et al (2019) | – |
| B4GALT5 1E10 | Steentoft et al (2019) | – |
| GNPTα | De Pace et al (2014) | – |
| LAMP2 2D5 | Radons et al (1992) | – |
| scFv M6P-1 | Müller-Loennies et al (2010) | – |

All commercially available antibodies used in this study are listed in Appendix Table S4.

| **Oligonucleotides and other sequence-based reagents** | | |
| --- | --- | --- |

All oligonucleotides used for Cas9 RNP-mediated genome editing and genotyping are listed in Appendix Table S1.

All oligonucleotides used for molecular cloning are listed in Appendix Table S2.

| **Chemicals, enzymes, and other reagents** | | |
| --- | --- | --- |
| TransIT-LT1 | Mirus | MIR 2300 |
| Cas9 | Integrated DNA Technologies | 1081058 |
| Neon electroporation kit | Thermo Fisher Scientific | MPK1025 |

| Reagent/resource | Reference or source | Identifier or catalog number |
|---|---|---|
| Quick-DNA microprep kit | Zymo Research | D3020 |
| DNA clean & concentrator kit | Zymo Research | D4014 |
| PureLink HiPure Plasmid Filter Midiprep Kit | Thermo Fisher Scientific | K210015 |
| Monarch Plasmid Miniprep Kit | New England Biolabs | T1010L |
| GeneJet Plasmid Miniprep kit | Thermo Fisher Scientific | K0502 |
| DirectPCR Lysis Reagent | Viagen Biotech | 102-T |
| FastDigest restrictions enzymes, DreamTaq | Thermo Fisher Scientific | - |
| Q5 polymerase | New England Biolabs | M0491S |
| Gibson Assembly kit | New England Biolabs | E5510S |
| Q5 Site-Directed Mutagenesis Kit | New England Biolabs | E0554S |
| X-tremegene 360 | Roche Life Sciences | XTG360-RO |
| Lipofectamin 3000 | Thermo Fisher Scientific | L3000008 |
| blasticidin | Invivogen | ant-bl-05 |
| hygromycin B gold | Invivogen | ant-hg-1 |
| puromycin | Invivogen | ant-pr-1 |
| Cumate solution | System Biosciences | QM150A-1 |
| Pierce™ BCA Protein Assay Kit | Thermo Fisher Scientific | 23227 |
| Micro-BCA™ Protein Assay-Kit | Thermo Fisher Scientific | 23235 |
| DC Protein Assay | Bio-Rad Laboratories | 5000112 |
| PageRuler Plus | Thermo Fisher Scientific | 26619 |
| Broad Range Prestained Protein Marker | Protein Tech Group | PL00002 |
| Protein A Dynabeads | Thermo Fisher Scientific | 10001D |
| anti-c-myc magnetic beads | Thermo Fisher Scientific | 88842 |
| anti-Flag M2 agarose beads | Sigma-Aldrich | A2220 |
| anti-HA agarose beads | Thermo Fisher Scientific | 26181 |
| Immobilon ECL Ultra | Merck | WBULS0100 |
| SuperSignal West Pico Plus | Thermo Fisher Scientific | 34577 |
| SuperSignal West Femto | Thermo Fisher Scientific | 34095 |
| SuperSignal West Atto | Thermo Fisher Scientific | A38554 |
| ECL prime | Cytiva | RPN2236 |
| Clarity Max | Bio-Rad Laboratories | 1705060 |

| Reagent/resource | Reference or source | Identifier or catalog number |
|---|---|---|
| Drugs/inhibitors used are listed in Appendix Table S3. | | |
| All other chemicals and the respective sources are listed in "Methods" section. | | |
| **Software** | | |
| CLC Main Workbench 8 | Qiagen | – |
| DECODR | Bloh et al (2021) | – |
| Adobe Photoshop | Adobe | – |
| Adobe Illustrator | Adobe | – |
| Image Lab | Bio-Rad Laboratories | – |
| R studio | – | – |
| AlphaFold 3 | Abramson et al (2024) | – |
| DeepTMHMM | Hallgren et al (2022) | – |
| UCSF ChimeraX v.1.7.1 | Meng et al (2023) | – |
| Colabfold v.1.5.5 | Mirdita et al (2022) | – |
| Spectronaut (version: 18.6.23) | Biognosys | – |
| Zen Black | Zeiss | – |
| Image J2 v2.14.0/1.54f; Build:c89e8500e4 | U.S. National Institutes of Health | – |
| **Other** | | |
| LSM880 confocal microscope (Airyscan) | Zeiss | – |
| JEM- 2100Plus Transmission Electron Microscope | Jeol | – |
| Neon electroporation device | Thermo Fisher Scientific | – |
| Amersham ImageQuant 800 | Cytiva | – |
| Chemidoc | Bio-Rad Laboratories | – |
| Typhoon | Cytiva | – |
| Dionex Ultimate 3000 nanoUHPLC system coupled to an Orbitrap Fusion Lumos | Thermo Fisher Scientific | – |
| Sonicator | Branson | – |

## Cell lines and cell culture

HEK293 and hTERT RPE-1 cells were from the American Type Culture Collection (ATCC), and 293FT cells were from Thermo Fisher Scientific. HAP1 cells as well as *LYSET* KO and *GNPTAB* KO HAP1 cells were from Horizon Discovery and were reported earlier (Richards et al, 2022). *B4GALT5/B4GALT6 KO* HEK293 cells (clone 1F6) were reported earlier (Narimatsu et al, 2019). Flp-In T-REx 293 cells were from Thermo Fisher Scientific. *LYSET* KO, *GNPTAB* KO (Richards et al, 2022) and *SPPL3* KO (Hobohm et al, 2022) HEK293 cells were reported previously. All cells were

cultivated under standard conditions and confirmed negative for mycoplasma contamination by routine PCR. HEK293 and Flp-In T-REx 293 cells were kept in DMEM GlutaMAX™ medium (Thermo Fisher Scientific) supplemented with 10% (v/v) fetal calf serum (PAN biotech or Serana), non-essential amino acids (Sigma-Aldrich or Thermo Fisher Scientific) and penicillin/streptomycin (Sigma-Aldrich). Medium for Flp-In T-REx 293 cells was also supplemented with 15 µg/ml blasticidin and 100 µg/ml zeocin (Invivogen), and tetracycline-free fetal calf serum (Capricorn Scientific) was used for these cells. HAP1 cells were kept in IMDM medium (Sigma-Aldrich or Thermo Fisher Scientific) supplemented with 10% (v/v) fetal calf serum, L-glutamine (Sigma-Aldrich), and penicillin/streptomycin. For hTert RPE-1 cells, DMEM/F12 medium containing HEPES and L-glutamine (Sigma-Aldrich) supplemented with 10% (v/v) fetal calf serum and penicillin/streptomycin was used.

## Lentiviral transduction

HEK293 cells that stably overexpress both LYSET-FLAG and GNPTAB-myc were generated using lentiviral transduction. Lentivirus was produced in 293FT cells by co-transfection of the transgene-expressing plasmid (GNPTAB-myc-IRES-LYSET-FLAG in pLenti_CMVTRE3G-Puro-DEST, see "Molecular cloning") and packaging plasmids ΔVPR, VSV-G and pAdvantage using TransIT-LT1 transfection reagent (Mirus, Cat# MIR 2300). The lentivirus-containing supernatant was harvested and filtered (0.45 µm). The lentivirus solution was then used to transduce the HEK293 Tet-On 3 G cell line (TaKaRa, Cat# 631182), which was supplemented with protamine sulfate at a final concentration of 8 µg/mL. Approximately 48 h post transduction, 2 µg/mL puromycin was added to select cells that had been successfully transduced.

## CRISPR/Cas9 genome editing

Genome editing was performed using Cas9 RNPs delivered by electroporation. Target-specific pre-designed crRNAs (Appendix Table S1) as well as generic tracrRNA and Cas9 protein were purchased from Integrated DNA Technologies. crRNA(s) and tracrRNA were mixed at a molar ratio of 1:1 (total crRNA:total tracrRNA). Cas9 loading was performed according to the manufacturer's instructions. $0.2 \times 10^6$ HEK293, 293FT, hTERT RPE-1 or HAP1 cells were electroporated with Cas9 RNP (30 pmol Cas9 & 36 pmol crRNA:tracrRNA duplex) in 10 µl buffer R using a Neon electroporation device (Thermo Fisher Scientific) with 2 pulses, each 1150 V and 20 ms width, for HEK293 cells, 2 pulses (1300 V and 20 ms width) for hTERT RPE-1 and 3 pulses (1575 V and 10 ms width) for HAP1 cells. *SPPL3* KO in Flp-In T-REx 293 cells was achieved using a multi-guide approach with reagents obtained from Synthego as detailed earlier (Hobohm et al, 2022). *LYSET* knock-in cells were generated as described earlier (Hobohm et al, 2022; Truberg et al, 2022) with the electroporation conditions detailed above and using a single-strand oligonucleotide (Integrated DNA Technologies). Following electroporation, cells were rested in 12-well plates for 72 h. They were then seeded on 96-well plates for single-cell cloning. Clones were expanded, genotyped and cryopreserved. Genomic DNA was isolated using a Quick-DNA microprep kit (Zymo Research) or DirectPCR lysis reagent (Viagen). Genomic regions flanking the editing sites were PCR-amplified with specific oligos (Appendix Table S1) using DreamTaq

(Thermo Fisher Scientific) polymerase. Detailed PCR reaction conditions can be made available upon request. PCR products were purified using DNA clean & concentrator kits (Zymo Research) and Sanger-sequenced using oligos also used for PCR amplification at Eurofins Genomics. Sequencing results were analyzed and visualized using CLC Main Workbench 8 (Qiagen). Sanger read deconvolution to assess genome editing results in heterozygous clones was performed using DECODR (Bloh et al, 2021).

## Molecular cloning

Restriction enzymes, PNK and T4 ligase were from Thermo Fisher Scientific or New England Biolabs. Inserts of all newly generated plasmids and expression constructs were confirmed by Sanger sequencing at Eurofins Genomics. Sequences of all oligonucleotides used for molecular cloning in this study are compiled in Appendix Table S2. Cloning of human SPPL3-V5 into the pENTR1A backbone and the SPPL3-V5 expression construct in pcDNA5/FRT/TO/DEST were described before (Hobohm et al, 2022). Site-directed mutagenesis was performed to introduce the D271N active site mutant into the SPPL3 coding sequence within the pENTR1A backbone. The strategy was designed using the NEBaseChanger (v1) online tool (New England Biolabs). Plasmid DNA was amplified using Q5 polymerase (New England Biolabs) and oligonucleotides listed in Appendix Table S2. Following amplification, the PCR reaction mix was treated with PNK and DpnI for 1.5 h at 37 °C in T4 ligase buffer. T4 ligase was added and the reaction mixture was incubated for 1 h at room temperature and subsequently heat-shock transformed into *E. coli* DH5a. After Sanger sequencing had confirmed correct introduction of the desired mutant, the SPPL3 D271N coding sequence was shuttled into pcDNA5/FRT/TO/DEST using LR clonase II (Thermo Fisher Scientific). Similarly, the GFP expression construct used was generated by shuttling GFP coding sequence from pENTR1A-GFP-N2 (FR1) (item #19364; Addgene) into pcDNA5/FRT/TO/DEST using LR clonase II. For the generation of GOLPH3 overexpression constructs, the coding sequence of human *GOLPH3* was obtained from Addgene (pEF-DEST51-GOLPH3, item #21690). Oligonucleotides were designed to amplify the *GOLPH3* coding sequence and replace the sequence coding for a C-terminal V5-His tag with a stop codon. In addition, HindIII and XhoI restriction enzyme cleavage sites were introduced upstream and downstream, respectively, of the *GOLPH3* coding sequence during PCR amplification with Q5 polymerase. The purified PCR product was cloned into pcDNA3.1 via HindIII and XhoI (Thermo Fisher Scientific). GOLPH3 point mutations were also introduced by side-directed mutagenesis as detailed above using oligonucleotides listed in Appendix Table S2. To enable cumate-inducible ectopic expression of GOLPH3 WT and its mutants, GOLPH3 coding sequence (with C-terminal tag) was again amplified by PCR and cloned into pBQM812A (System Biosciences) via NheI/NotI. Coding sequences for LYSET isoforms 1 and 2 were amplified from previously described constructs (Richards et al, 2022) and inserted into pcDNA6 myc/His via HindIII and BamHI restriction sites. LYSET mutants were generated as outlined above or using a Q5 side-directed mutagenesis kit (New England Biolabs) following the manufacturer's instructions. To generate a B4GALT1-V5 expression construct, cDNA encoding B4GALT1 (corresponding in sequence to NP_001488.2) was PCR-amplified from Str-KDEL-

flGALT-SBP-tagBFP (Addgene item #65274) and cloned via HindIII and BamHI sites into a previously described pcDNA3.1-derived plasmid containing coding sequence for a C-terminal V5 tag (Hobohm et al, 2022). Similarly, a B4GALT5-V5 expression construct was cloned following PCR amplification of B4GALT5 coding sequence (corresponding to RefSeq ID NM_004776) kindly provided by Marco Trinchera (Insubria University, Italy). To implement synchronized B4GALT5 trafficking using the RUSH system, B4GALT5 coding sequences was PCR-amplified and cloned into Str-KDEL_ST-SBP-mCherry (Addgene item #65265) using PteI and EcoRI. The GNPTAB-myc-IRES-LYSET-FLAG expression construct for lentivirus transduction was generated using Gibson Assembly (New England Biolabs). The coding sequence for *GNPTAB* was PCR-amplified from pcDNA3.1-GNPTAB (item # 78108; Addgene), and the myc-IRES-LYSET-FLAG-containing fragment was synthesized at IDT. Both fragments were assembled into the pLenti_CMVTRE3G-Puro-DEST (w1118-1) vector (item # 27565; Addgene) and digested with EcoRV using Gibson Assembly.

## Cell culture experiments, plasmid transfections, and generation of stable cell lines

For some cell culture experiments, HEK293 and Flp-In T-REx 293 cells were plated on poly-L-Lys (Sigma-Aldrich)-coated cell culture plastics. All other cell lines were plated on uncoated cell culture plastics. Cells were plated in the corresponding cell culture medium lacking selection antibiotics, were grown to confluency and then harvested (see below). To collect conditioned cell culture supernatants, cells were washed with PBS and subsequently warm OptiMEM GlutaMAX (Thermo Fisher Scientific) was added. Before analysis, cells were cultivated for another 16 to 24 h in OptiMEM GlutaMAX at 37 °C. Transient transfections in HEK293 cells were performed using X-tremegene 360 (Roche Life Sciences), TransIT-LT1 (Mirus Bio) or Lipofectamine 3000 (Thermo Fisher Scientific) transfection reagents following the manufacturers' instructions. For treatment with small-molecule inhibitors, these compounds (listed in Appendix Table S3) were added to cell culture media and cells there then cultivated for another 16–24 h at 37 °C. Protease inhibitor treatments for subcellular fractionation and immuno-fluorescence microscopy experiments were performed for 24 h. Stable re-introduction of SPPL3 into *SPPL3* KO Flp-In T-REx 293 cells was achieved following transfection with the Flp recombinase-encoding plasmid pOG44 (Thermo Fisher Scientific) and the corresponding SPPL3-encoding pcDNA5/FRT/TO plasmid (see above) at a ratio of 9:1 using the TransIT-LT1 reagent. Two days after transfection, the selection regimen was changed to 10 µg/ml blasticidin and 100 µg/ml hygromycin B gold (Invivogen) and resistant clones were expanded. SPPL3 expression in these cells was induced by supplementing cell culture media with doxycycline (Sigma-Aldrich) for 24–48 h. For the B4GALT5 RUSH experiment, cells kept in medium supplemented with dialyzed FCS (Capricorn Scientific) were transiently transfected with Str-KDEL_B4GALT5-SBP-mCherry using TransIT-LT1. Twenty-four hours after transfection, cells were washed twice with prewarmed PBS. Serum-free DMEM with or without 50 µM D-biotin (Carl Roth) was added and cells were kept at 37 °C as indicated. To establish *GOLPH3/GOLPH3L*-deficient HEK293 cells with a cumate-inducible re-introduction of GOLPH3 or mutants thereof, clone VF3 was

transfected with the pBQM812A plasmid containing the corresponding GOLPH3 coding sequence along with an expression plasmid for a hyperactive PiggyBac transposase (ratio of both plasmids 5:2) using TransIT-LT1. One day after transfection, cells were expanded in fresh medium. At day three after transfection, cells were diluted into fresh culture medium supplemented with 1 µg/ml puromycin (Invivogen) for selection of pools of stable transfectants. These cells were kept under puromycin selection, expanded, cryoconserved and used for experiments. GOLPH3 expression was induced by supplementing media with 1× cumate (diluted from a 10,000× water-soluble cumate solution, System Biosciences).

## Preparation of whole-cell and membrane lysates

To harvest cells, culture medium was removed and plates were washed once with ice-cold PBS. Cells were then gently scraped off the cell culture dish in 1 ml of ice-cold PBS and collected by centrifugation $(500 \times g,\ 4\,°C,\ 5\,\text{min})$. The supernatant was completely removed, and pellets were either directly used or frozen at $-20\,°C$. Whole-cell lysates were obtained by resuspending cell pellets in ice-cold RIPA buffer (150 mM NaCl, 50 mM Tris HCL (pH 7.4), 1 mM EDTA, 0.5% (w/v) sodium deoxycholate, 0.1% (w/v) SDS, 1% (v/v) Triton X-100 and protease inhibitor mix (Roche)). Following incubation on ice, lysates were cleared by centrifugation $(21,000 \times g,\ 30\,\text{min},\ 4\,°C)$. For membrane isolation, cells were resuspended in ice-cold hypotonic buffer (10 mM Tris (pH 7.6), 1 mM EDTA, 1 mM EGTA, pH 7.6) and sheered using a 23 G needle. Following sedimentation of nuclei $(2400 \times g,\ 5\,\text{min},\ 4\,°C)$, membrane pellets were obtained by centrifugation at $100,000 \times g$. Membrane pellets were washed twice with carbonate buffer (0.1 M $Na_2CO_3$, 1 mM EDTA, pH 11.3), once with STE buffer (150 mM NaCl, 50 mM Tris, 2 mM EDTA, pH 7.6), and then lysed in RIPA. The total protein content of whole-cell lysates and membrane lysates was determined using BCA and Micro-BCA kits, respectively (both from Thermo Fisher Scientific). Samples were mixed with 4× LDS sample buffer, supplemented with DTT (final concentration 50 mM) and subjected to SDS-PAGE.

## Enrichment of proteins secreted into conditioned cell culture media

Conditioned OptiMEM medium was collected from cell culture dishes and immediately centrifuged for 5 min at $500 \times g$ and 4 °C to remove detached cells. The supernatant was transferred to ultracentrifuge tubes (Beckman Coulter) and subjected to $100,000 \times g$ centrifugation to remove exosomes. The supernatant was carefully removed and frozen. Total protein precipitation from conditioned supernatants was performed using trichloroacetic acid (TCA). Thawed conditioned supernatants were mixed with freshly prepared aqueous solution of 100% (w/v) TCA (Carl Roth or Sigma-Aldrich) to reach a final TCA concentration of 20% (w/v). Solutions were vortexed extensively and incubated on ice for one hour. Following centrifugation at $21,000 \times g$ for 45 min, the supernatant was removed and precipitates were washed twice with acetone ($-20\,°C$). Following acetone evaporation, precipitated proteins were resuspended in 2× LDS sample buffer with 100 mM DTT. For enrichment and subsequent deglycosylation of secreted

proteins, collected culture supernatants were centrifuged at $500 \times g$ and 4 °C for 5 min. The supernatant was then transferred to Amicon Ultra-4 centrifugal filter units (10 kDa MW cut-off). It was concentrated roughly 40-fold by centrifugation at $2000 \times g$ and 4 °C. SDS (final concentration: 1% (w/v)) was added to the concentrated supernatant, and proteins were denatured at 95 °C for 10 min. Subsequently, the solution was mixed with deglycosylation buffer (50 mM $Na_2PO_4$, pH 7.2, 12 mM EDTA, 0.4% (v/v) NP-40) to adjust the SDS concentration to 0.2% (w/v). In total, 1 µl PNGase F (Roche) was added and the reactions were incubated at 37 °C overnight and stopped by the addition of sample buffer. Control samples were incubated overnight without the addition of glycosidase.

## Co-immunoprecipitations of ectopically expressed GOLPH3 and LYSET

Immediately after harvesting, PBS-washed cells were lysed in an appropriate amount of lysis buffer (50 mM Tris pH 7.6, 150 mM NaCl, 2 mM EDTA, 0.5% (v/v) Igepal CA-630 (Sigma-Aldrich), freshly supplemented with 1× Protease Inhibitor Cocktail (Roche)) for 30 min on ice. Lysates were cleared by centrifugation at $21,000 \times g$ and 4 °C for 20 min. Protein content was quantitated using a BCA assay (Thermo Fischer Scientific). In all, 200 µg total protein was brought to a total volume of 1 ml with lysis buffer and agitated at 4 °C overnight in the presence of 1 µl GOLPH3 antibody (Protein Tech Group #19112-1-AP or Abcam #ab98023) and 30 µl Protein A Dynabeads equilibrated in lysis buffer (Thermo Fisher Scientific). The next day, the samples were washed four times with lysis buffer and resuspended in 2× LDS sample buffer supplemented with 100 mM DTT. Samples were denatured for 10 min at 70 °C and analyzed by immunoblotting.

## Immunoblotting

Unless described otherwise, immunoblotting was performed as follows: Samples containing LDS sample buffer supplemented with DTT were denatured (10 min at 70 °C in most cases; 30 min at 37 °C in case of SPPL3 immunoblots), separated on user-cast Bis-Tris gels using 1× MOPS buffer (50 mM MOPS, 50 mM Tris, 0.1% (w/v) SDS, 1 mM EDTA, pH 7.7) and transferred to methanol-activated PVDF membranes with 0.45 µm pore width (Merck) by tank blotting in 25 mM Tris, 192 mM glycine, pH 8.3 supplemented with 10% (v/v) methanol. Blocking and antibody incubation steps were done in 5% (w/v) non-fat dry milk (Carl Roth) dissolved in TBS supplemented with 0.2% (v/v) Tween-20 (TBS-T), with the exception of anti-SPPL3 immunostainings which were done using 5% (w/v) bovine serum albumin (Carl Roth) dissolved in TBS-T. All primary (Appendix Table S4) and secondary antibodies (purchased from Dianova, Cytiva or R&D Systems) were diluted in blocking solution and incubation steps were performed overnight at 4 °C and for 1 to 2 h at room temperature, respectively. For immunoblot detection using crude hybridoma supernatants, PVDF membranes were blocked with 5% (w/v) non-fat dry milk dissolved in TBS-T and incubated in hybridoma supernatant diluted with blocking buffer (1:1). Antibody binding was visualized using ECL reagents purchased from Merck (Immobilon ECL Ultra), Thermo Fisher Scientific (SuperSignal West Pico Plus, Femto or Atto), Cytiva (ECL prime) or Bio-Rad Laboratories (Clarity Max) and

signals were acquired using an Amersham ImageQuant 800 (Cytiva) or Chemidoc (Bio-Rad Laboratories) system. Image cropping and adjustment was conducted using Adobe Photoshop or Image Lab (Bio-Rad Laboratories). Total protein stain on PVDF membranes was done using the No-Stain protein labeling reagent (Thermo Fisher Scientific) and signals (570/20 bandpass filter) were documented using the Typhoon scanner following dye excitation at 488 nm.

## Subcellular fractionation

Subcellular fractionation was performed as previously described (Richards et al, 2022). Briefly, HAP1 or HEK293 cells were washed three times with ice-cold PBS and once with 250 mM Sucrose 10 mM HEPES pH 7.4, harvested by centrifugation at $500 \times g$ for 5 min, and homogenized by 25 strokes in a tight-fitting 7 ml Douncer (Wheaton) in 2 ml in homogenization buffer (250 mM sucrose, 1 mM EDTA, 10 mM HEPES, pH 7.4). Nuclei were pelleted by 10 min centrifugation at $800 \times g$ and 4 °C. The post-nuclear supernatant (PNS) was recovered, the nuclear pellet re-homogenized using 15 strokes in 1 mL homogenization buffer and centrifuged as above. The PNS were pooled, and the lysosomes were pelleted by centrifugation at $20,000 \times g$ for 30 min and washed once with homogenization buffer (20k fraction). The $20,000 \times g$ supernatants were centrifuged at $100,000 \times g$ for 1 h in a Sorvall Discovery M120 ultracentrifuge to obtain the 100k Golgi-enriched fraction. Both 20k and 100k pellets were solubilized in RIPA buffer (Thermo Fisher Scientific) containing complete protease inhibitors (Roche), lysed on ice for 20 min and cleared by centrifugation at $17,000 \times g$ for 20 min. Protein concentrations were determined using the DC Protein Assay (Bio-Rad Laboratories). Equal amounts of protein were solubilized in LDS sample buffer supplemented with 100 mM DTT (final concentration), incubated at 70 °C for 10 min, and run on 4–12% NuPAGE Bis-Tris gels (Thermo Fisher Scientific) using 1× MES running buffer (Thermo Fisher Scientific). Blotting was performed using tank transfer in Bis-Tris transfer buffer (Thermo Fisher Scientific) containing 10% MeOH and nitrocellulose (for detection of the GNPTα subunit) or RTA transfer stacks (Bio-Rad Laboratories). Cells were treated with protease inhibitors (5 µg/ mL E64d (Sigma-Aldrich), 100 µM Leupeptin (Sigma-Aldrich), 100 µM Pepstatin A (Roche)) in growth medium for 24 h where indicated.

## Affinity purification of M6P-tagged proteins

HEK293 or HAP1 cells were washed three times with PBS and incubated in OptiMEM containing 10 mM $NH_4Cl$ for 24 h. Conditioned media were collected, detached cells were pelleted by centrifugation at $500 \times g$ for 10 min, and the supernatants concentrated 10x using Amicon Ultra-15 centrifugal filter units (10 kDa cut-off, Millipore). The concentrate was adjusted to 10 mM Tris pH 7.4 and 0.2% (w/v) Triton X-100. In all, 3 µg of scFv M6P-1 (Müller-Loennies et al, 2010) and 30 µl myc magnetic beads (Thermo Fisher Scientific) washed with PBS containing 1 mM EDTA were used to precipitate M6P-tagged proteins at 4 °C for 3 h. Beads were washed three times for 5 min with 500 µl (PBS with 0.2% (w/v) Triton X-100 and complete protease inhibitors), three times for 5 min with 500 µl 10 mM glucose 6-phosphate, 10 mM

mannose in PBS containing 0.2% (w/v) Triton X-100 containing complete protease inhibitors, and three times for 5 min with 500 μl 10 mM NaPi (pH 7.4). M6P-tagged proteins were eluted in 2×10 min using 100 μl 10 mM Disodium M6P in 10 mM NaPi (pH 7.4) containing complete protease inhibitors. All steps were performed at 4 °C and on ice, respectively. Overall, 30 μl of the eluates and concentrated media corresponding to 5% of the input were used for immunoblotting using 10% Tris-Glycine gels. For M6P detection, blocking and antibody incubations were performed using 3% (w/v) receptor grade BSA in TBS-T as described previously (Richards et al, 2022).

## FLAG-immunoprecipitation of LYSET complexes

Confluent HEK293 GNPT-myc-IRES-LYSET-FLAG cells grown on 100 mm culture dishes were treated with 1 μg/mL Doxycycline (Clontech) for 24 h in DMEM supplemented with GlutaMAX (Gibco), 1% penicillin/streptomycin (P/S) and 10% tetracycline-free FBS (tet-FBS; Clontech). After removal of the medium, the cells were washed and harvested and lysed in lysis buffer, (25 mM Tris/HCl pH 7.4, 150 mM NaCl containing 1 mM EDTA, 5% (v/v) glycerol, containing 30 mM n-octylglucopyranoside (OGP) detergent (Sigma-Aldrich), protease inhibitors (Sigma-Aldrich) and PhosStop (Roche) for 30 min on ice, gently vortexing every 10 min. Cell lysates were then centrifuged at $100,000 \times g$ for 15 min in a Sorvall Discovery M120 ultracentrifuge. The supernatant was collected, and the remaining membrane pellet re-extracted in the same volume of lysis buffer for 30 min on ice followed by centrifugation at $100,000 \times g$ for 15 min. Both supernatants were combined and processed for a preabsorption using TBS-washed anti-HA agarose beads (Thermo Fisher) for 2 h on a rotating wheel at 4 °C. The beads were separated by centrifugation. An aliquot of the clarified lysates was saved as an input fraction, while the rest was incubated under rotation overnight at 4 °C with anti-FLAG M2 agarose beads (Sigma-Aldrich). The beads were separated from the non-bound fraction (NB) by centrifugation. The beads were then washed five times with TBS containing 15 mM OGP (TBS-OGP). For the elution of LYSET-FLAG and specifically LYSET interacting proteins the beads were incubated with 250 μl using the FLAG® peptide (Thermo Fisher) at a final concentration of 150 ng/μL TBS-OGP for 30 min on a rotating wheel at 4 °C followed by centrifugation at $5000 \times g$ for 1 min (E1). The elution was repeated four more times (E2-E5). Samples were solubilized using 4X LDS sample buffer (Thermo Fisher) supplemented with 400 mM DTT, separated on NuPAGE 4–12% Bis-Tris Protein Gels (Thermo Fisher), and transferred by tank blotting to nitrocellulose membranes using Tris-Glycine transfer buffer supplemented with 20% (v/v) methanol. Blocking was performed with 5% non-fat milk in TBS containing 0.05% Tween-20 (TBS-T) except for anti-FLAG blots, where 5% receptor grade BSA (Serva) in TBS-T was used. Membranes were incubated overnight at 4 °C with the following antibody dilutions in blocking buffer: rabbit anti-TMEM251 (LYSET) (1:1000; Invitrogen, PA5-61769), rat anti-GNPT α-subunit hybridoma supernatant (1:50) (De Pace et al, 2014), mouse anti-FLAG (1:1000; Sigma-Aldrich, F1804), mouse anti-MYC (1:1000; Cell Signaling Technology, 2276), rabbit anti-GOLPH3 (1:1000; Abcam, 230809).

Horseradish peroxidase (HRP)-coupled anti-rabbit IgG (Biozol, JIM-111-035-003), anti-mouse IgG (Jackson, 115-035-003), or anti-

rat IgG (Jackson,112-035-003) were used as secondary antibodies and incubated in blocking buffer for 1 h at room temperature at a dilution of 1:5000. Immunoreactive proteins were detected by enhanced chemiluminescence (Clarity Western ECL substrate kit; Bio-Rad, 1705060) and imaging in a Chemidoc Imaging system (Bio-Rad).

## Immunofluorescence stainings

HAP1 cells were fixed using 4% (w/v) PFA in PBS (Santa Cruz) for 12 min at room temperature, washed once with 100 mM Glycine in PBS and two times with PBS. Blocking, permeabilization and antibody incubations were done in 3% (w/v) BSA, 0.2% (w/v) saponin in PBS and washed in PBS. Commercial primary antibodies are listed in Appendix Table S4 and rat anti-GNPTα (1:100) as well as mouse anti-LAMP2 (clone 2D5, 1:200) were reported previously (De Pace et al, 2014; Radons et al, 1992). Alexa Fluor Plus 488 or 594-coupled secondary antibodies and Hoechst 33342 (all Thermo Fisher Scientific) were diluted 1:1000. Coverslips were mounted using Aqua/Polymount (Polysciences) and 16-bit images acquired using a Plan-Apochromat 63×/1.4 Oil DIC M27 oil immersion objective on a LSM880 confocal microscope equipped with an Airyscan detector in SR-mode (Zeiss). Images were acquired, and Airyscan processing was performed using Zen Black (Zeiss). Image J2 (v2.14.0/1.54f; Build:c89e8500e4) was used for adjusting maximal and minimal thresholds and merging single-channel images. Thresholds were equally set for all images analyzed in each experiment. Cells were treated with protease inhibitors (5 μg/mL E64d (Sigma-Aldrich), 100 μM Leupeptin (Sigma-Aldrich), 100 μM Pepstatin A (Roche)) in growth medium for 24 h where indicated.

## Electron microscopy

HAP1 cells were fixed with 1% (w/v) glutaraldehyde and 4% (w/v) paraformaldehyde in 0.1 M phosphate buffer overnight. Thereafter, they were harvested and spun down at $1000 \times g$ for 10 min in a 1.5 mL microcentrifuge tube. The cells were spun down three times after each washing step with 0.1 M cacodylate buffer and embedded in 3% (w/v) agarose. The solidified agarose-containing pellet was cut on ice into small pieces of about 1–2 mm³. After three more washing steps, the tissue blocks were prepared as described previously (West et al, 2010). Briefly, blocks were post-fixed in 0.1% (w/v) potassium ferrocyanide-reduced 2% (w/v) osmium tetroxide in cacodylate buffer for 1 h. Thereafter, the blocks were rinsed in distilled water and treated with 0.1% aqueous thiocarbohydrazide, which stains carbohydrates, for 20 min. After further rinsing in distilled water, the tissue was liganded with 2% (w/v) osmium tetroxide again for 30 min before staining with uranyl acetate and lead aspartate. The pieces were further dehydrated using ascending ethyl alcohol concentration steps, followed by two rinses in propylene oxide. Infiltration of the embedding medium was performed by immersing the samples in a 1:1 mixture of propylene oxide and Epon and finally in neat Epon and polymerized at 60 °C. Semithin sections (0.5 μm) were prepared for light microscopy, mounted on glass slides and stained with 1% Toluidine blue. Images were acquired with a JEM- 2100Plus Transmission Electron Microscope at 200 kV (Jeol) equipped with a XAROSA CMOS camera (Emsis).

## Mass spectrometry sample preparation

Parental and GOLPH3/GOLPH3L KO HEK293 cells (from four separately frozen passages) were thawed and seeded onto uncoated 15 cm plastic dishes. After three washes with PBS, cells were incubated with OptiMEM (Thermo Fisher Scientific) for 24 h. The medium was collected, centrifuged 10 min at $500 \times g$ to pellet detached cells and the supernatant concentrated 10x using Amicon-15 centrifugal filter units (cut-off 10 kDa, Millipore). Cells were washed 3× with ice-cold PBS, harvested, pelleted at $500 \times g$ and 4 °C for 10 min, and snap-frozen. Cell pellets were resuspended in 700 μL lysis buffer (4% SDS, 100 mM HEPES pH 8.0) and incubated at 95 °C for 10 min at 1000 rpm in a thermomixer. Subsequently, the samples were lysed using a sonicator (Branson) with three cycles of 30 s each at 4 °C. Cell lysates were centrifuged at $20,000 \times g$ for 30 min at room temperature (RT) and the clear supernatants were transferred to new microtubes. The protein concentration of both cell lysates and concentrated media samples was determined using the DC protein assay (Bio-Rad Laboratories) and 150 μg of protein of each sample (both cell lysates and media) was precipitated by ice-cold acetone. The protein pellet was resuspended in 0.5% RapiGest (Waters), 100 mM $NH_4HCO_3$, pH 7.8 (Mosen et al, 2021). Proteins were reduced with DTT (final concentration of 5 mM) at 56 °C for 45 min, alkylated with 20 mM acrylamide (final concentration) for 30 min at RT, and the reaction quenched by adding 5 mM DTT (Müller and Winter, 2017). Samples were diluted with 100 mM $NH_4HCO_3$ to a final concentration of 0.1% RapiGest and proteins digested with sequencing-grade modified trypsin (Promega) at an enzyme/protein ratio of 1:100 (w/w) at 37 °C overnight. The next day, RapiGest was hydrolyzed by adding 1% trifluoroacetic acid (TFA) (final concentration) and incubation at 37 °C, 800 rpm for 30 min. Samples were centrifuged at $20,000 \times g$ for 10 min, and the clear supernatants were transferred to new microtubes. Peptides were desalted with Oasis HLB cartridges (Waters) and dried in a vacuum centrifuge. Dried peptides were resuspended in 5% acetonitrile (ACN), and the peptide concentration was determined using the quantitative fluorometric peptide assay (Thermo Fisher Scientific). Finally, peptides were dried again and resuspended in 5% ACN/5% formic acid (FA).

## LC-MS/MS analysis

Peptides were analyzed using a Dionex Ultimate 3000 nanoUHPLC system coupled to an Orbitrap Fusion Lumos mass spectrometer (Thermo Fisher Scientific). Peptide separation was performed using a 40 cm × 100 μm (inner diameter) in-house produced spray tip generated with a P-2000 laser puller (Sutter Instrument, Novato, CA, USA) packed with 3-μm Reprosil C18 AQ particles (Dr. Maisch) as analytical column. After loading of peptides directly to the analytical column at a flow rate of 850 nl/min with 100% solvent A (0.1% FA in water), they were separated with a 165 min linear gradient from 5% to 35% solvent B (95% ACN/0.1% FA). Eluting peptides were ionized in the positive ion mode and mass spectrometry (MS) analyses were performed in the data-independent acquisition (DIA) mode. All scans were acquired in the Orbitrap part of the instrument, MS1 scans at a resolution of 120,000, a mass range from $m/z$ 350 to 1200, and a maximum

injection time of 20 ms; and MS2 scans at a resolution of 30,000, a dynamic mass range, and a maximum injection time of 60 ms.

## Data analysis and visualization

For library generation, DIA *.raw files were analyzed with the Pulsar search engine available in Spectronaut (version: 18.6.23). Data were searched against UniProt Homo sapiens (entries: 20,375, release date: 01/2022) in combination with the cRAP database containing common contaminants. The following parameters were defined: enzyme: trypsin (except proline was the next amino acid); number of allowed missed cleavage sites: 2; minimum peptide length: 5 amino acids; mass tolerance: dynamic; fixed modification: propionamide at cysteine; variable modifications: oxidation at methionine and acetylation at protein N-terminus. The high-precision indexed retention time (iRT) concept (dynamic) was applied for retention time alignment. Mass tolerances for matching of precursors and fragment ions as well as peak extraction windows were determined automatically by Spectronaut. Data were normalized based on median abundance, and the results were filtered with 1% FDR on the precursor and protein level ($q$ value < 0.01). For the determination of $P$ values, the post-analysis pipeline of Spectronaut was used with default parameters (differential abundance testing: unpaired $t$ test). Volcano plots were generated using the ggplot2 package in R. Datasets were filtered for type II membrane proteins based on annotations in Uniprot as detailed earlier (Hobohm et al, 2022). Filtering for Golgi membrane proteins was done based on GO term annotations with manual corrections. Soluble lysosomal proteins were identified based on a previously reported manual selection of confidently characterized lysosomal enzymes (Markmann et al, 2017).

## Structural predictions

The structure of binary LYSET:GOLPH3 and LYSET:GOLPH3L protein complexes was predicted using ColabFold v1.5.5 (Mirdita et al, 2022) based on AlphaFold2 using MMseqs2 (Jumper et al, 2021) and AlphaFold2 Multimer (Evans et al, 2022). In case of GOLPH3, the experimentally determined structure (PDB ID3KN1, Wood et al, 2009) was matched to the monomeric structure of the AlphaFold2 prediction to control for proper folding. In case of GOLPH3L and LYSET where such experimental data were missing predictions of AlphaFold2 Multimer on ColabFold were compared to the AlphaFold Structure Database (Varadi et al, 2022) and ColabFold v.1.5.5 predictions based on RoseTTAFold2 (Baek et al, 2023) and OmegaFold (Wu et al, 2022). The best resulting protein complex structure was visualized in UCSF ChimeraX v1.7.1 (Meng et al, 2023). The pLDDT score, electrostatic potential, and transmembrane domains predicted by DeepTMHMM v1.0.24 (Hallgren et al, 2022) were used to apply the different coloring. The structures of ternary GNPTAB:GOLPH3:LYSET complexes were predicted using AlphaFold 3 (Abramson et al, 2024) through the official AlphaFold 3 server. The seed was set to auto and the main UniProt isoform sequences for human GNPTAB and GOLPH3 (or GOLPH3L) as well as the corresponding sequences for human LYSET isoform 1 and isoform 2 were used. Predicted complexes were visualized as detailed above.

### Re-analysis of published proteome data

Proteome data published for *Lyset* KO MEFs (Richards et al, 2022) and *GOLPH3/GOLPH3L* KO U2-OS cells (Welch et al, 2021) were retrieved from the online supplements. In R, these datasets were filtered for proteins annotated as type II membrane proteins in the human reference proteome in Uniprot as described earlier (Hobohm et al, 2022). In addition, datasets were also filtered for proteins confidently considered SPPL3 substrates (as defined in (Hobohm et al, 2022)). Filtered datasets were then plotted using the ggplot2 package in R.

## Data availability

Proteome data were deposited on the proteomeXchange server (project accession: PXD052083). Code in R used to analyze and visualize proteome data can be made available upon request. Newly generated reagents can be made available upon request.

The source data of this paper are collected in the following database record: biostudies:S-SCDT-10_1038-S44318-024-00305-z.

## Peer review information

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

## Acknowledgements

The authors would like to thank Elmar Wolf, Stefan Rose-John, Christoph Becker-Pauly, Paul Saftig and Markus Damme (Institute of Biochemistry, Kiel University) for discussions and support. Anna Hofmann and Antonia Zabel (Institute of Biochemistry, Kiel University) generated expression constructs. Expert technical assistance was provided by Stefanie Schnell, Michelle Stammermann and Kendra Bednarski (Institute of Biochemistry, Kiel University) as well as Chudamani Raithore, Emanuela Szpotowicz (both ZMNH, Hamburg) and Christina Nimke (IKMB, Kiel University). Marco Trinchera (Insubria University, Italy), Ulla Mandel (University of Copenhagen, Denmark), Florian Bleibaum and Sönke Rudnik (both Kiel, Germany) kindly provided reagents. This work was funded by the Deutsche Forschungsgemeinschaft (DFG) CRC877 (to BB and MV); DFG FOR2625 TP7 (to EC, DW, and TB), DFG RTG2771 (to EC and TB), the Fritz Thyssen Stiftung (to ZC and SJ), the China Scholarship Council (to JL), the Novo Nordisk Fonden NNF24OC0088218 (to YN) and the Faculty of Medicine of the Christian-Albrechts-Universität (CAU) zu Kiel (to SJ and MV).

## Author contributions

**Berit K Brauer**: Investigation; Visualization; Methodology. **Zilei Chen**: Investigation; Methodology. **Felix Beirow**: Investigation. **Jiaran Li**: Data curation; Formal analysis. **Daniel Meisinger**: Software; Formal analysis; Investigation; Visualization; Methodology. **Emanuela Capriotti**: Investigation; Methodology. **Michaela Schweizer**: Investigation. **Lea Wagner**: Methodology. **Jascha Wienberg**: Methodology. **Laura Hobohm**: Methodology. **Lukas Blume**: Methodology. **Wenjie Qiao**: Resources; Methodology. **Yoshiki Narimatsu**: Resources. **Jan E Carette**: Resources; Supervision; Funding acquisition. **Henrik Clausen**: Resources; Supervision; Funding acquisition. **Dominic Winter**: Data curation; Formal analysis; Supervision; Funding acquisition. **Thomas Braulke**: Conceptualization; Supervision; Funding acquisition; Writing—review and editing. **Sabrina Jabs**: Conceptualization; Formal analysis; Supervision; Funding acquisition; Investigation; Visualization; Methodology; Writing—original draft; Project administration; Writing—review and editing. **Matthias Voss**: Conceptualization; Data curation; Software; Formal analysis; Supervision; Funding acquisition; Investigation; Visualization; Methodology; Writing—original draft; Project administration; Writing—review and editing.

Source data underlying figure panels in this paper may have individual authorship assigned. Where available, figure panel/source data authorship is listed in the following database record: biostudies:S-SCDT-10_1038-S44318-024-00305-z.

## Funding

## Disclosure and competing interests statement

The authors declare no competing interests. YN and HC have a financial interest in GlycoDisplay Aps. HC has a financial interest in GO Therapeutics Inc. YN and HC's interests are reviewed and managed by the University of Copenhagen in accordance with their conflict of interest policies.

# Expanded View Figures

**Figure EV1.  Alphafold2 Multimer prediction of a potential interaction of LYSET and GOLPH3.**

(**A**) Schematic overview of the topology of LYSET. Based on Uniprot annotations, LYSET is a double-pass membrane protein with cytosolic N- and C-termini. Putative initiator methionines for the long and short isoform are highlighted in magenta and positively charged lysine and arginine residues in green. The sketch was generated using Protter (Omasits et al, 2014). (**B**) Alphafold2 Multimer prediction of a potential interaction of LYSET and GOLPH3 with coloring based on the prediction confidence score. (**C**) LYSET:GOLPH3 complex as depicted in (**B**) with GOLPH3 colored in red and LYSET in blue. Annotated TMD1 and TMD1 along with a putative TMD predicted by DeepTMHMM (Hallgren et al, 2022) in LYSET are colored in yellow. Insert, magnification of the LYSET:GOLPH3 interaction interface with side chains of select positively charged residues in LYSET and negatively charged residues in GOLPH3 shown. (**D**) Face-down view from the membrane plane on the predicted interaction of the LYSET N-terminus (blue) and GOLPH3 (surface charge shown). Side chains of positively charged amino acid residues in the LYSET N-terminus are shown. A similar putative cargo binding site is conserved in GOLPH3L (right). (**E**) Co-immunoprecipitation of LYSET and GOLPH3 in HEK293 transfectants. WT HEK293 cells were transfected as indicated and GOLPH3 was immunoprecipitated with anti-GOLPH3 (ab98023) from obtained cell lysates. Immunoprecipitates were analyzed by immunoblotting. HC, antibody heavy chain. (**F**) Transient overexpression of C-terminally myc-tagged LYSET (isoform 1) variants, that carry point mutations in select arginine residues in the N-terminal cytosolic region, in *LYSET* KO HEK293FT cells. LYSET protein levels were analyzed by immunoblotting with equal proteins amounts loaded and actin serving as a loading control. Source data are available online for this figure.

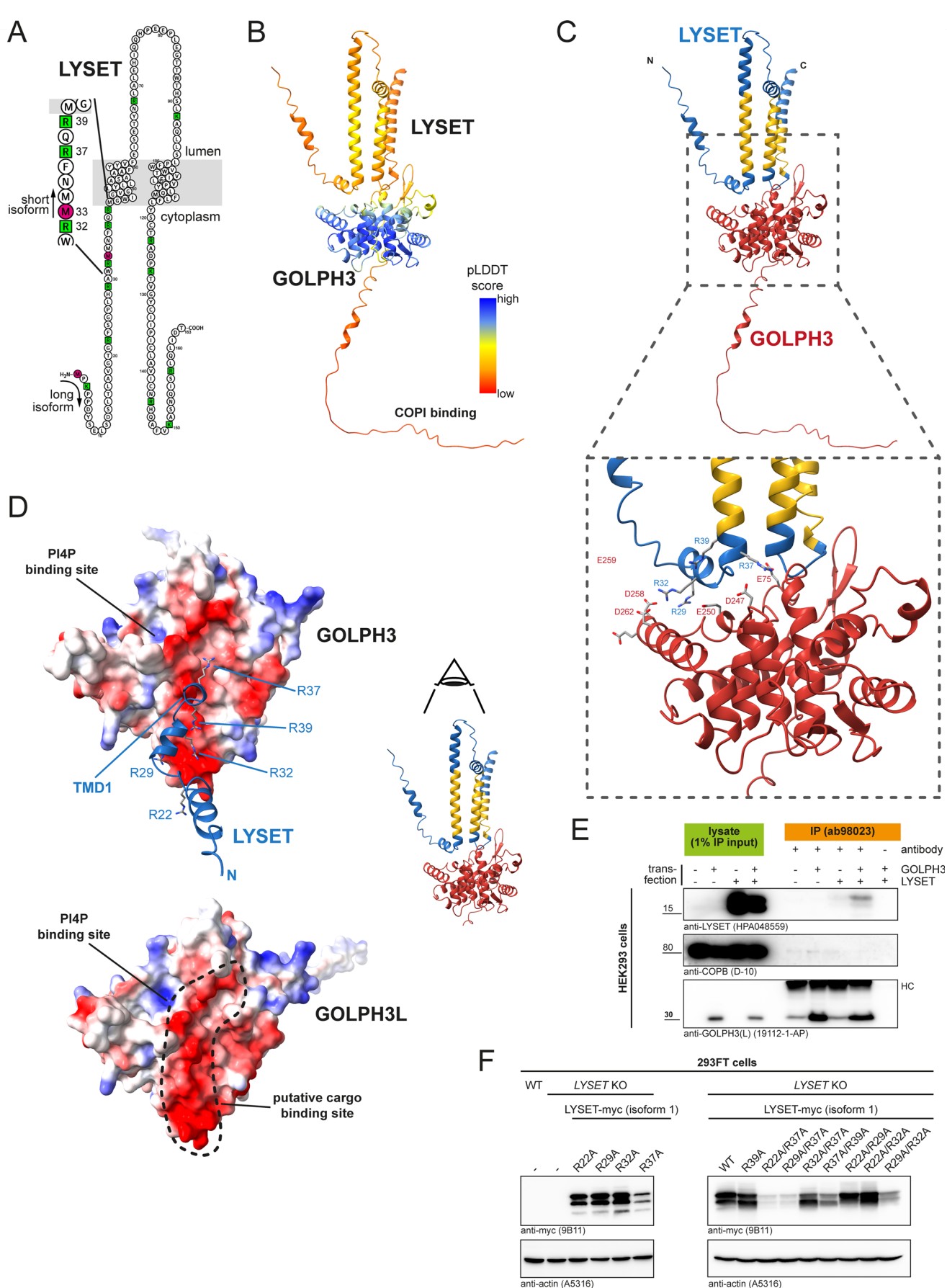

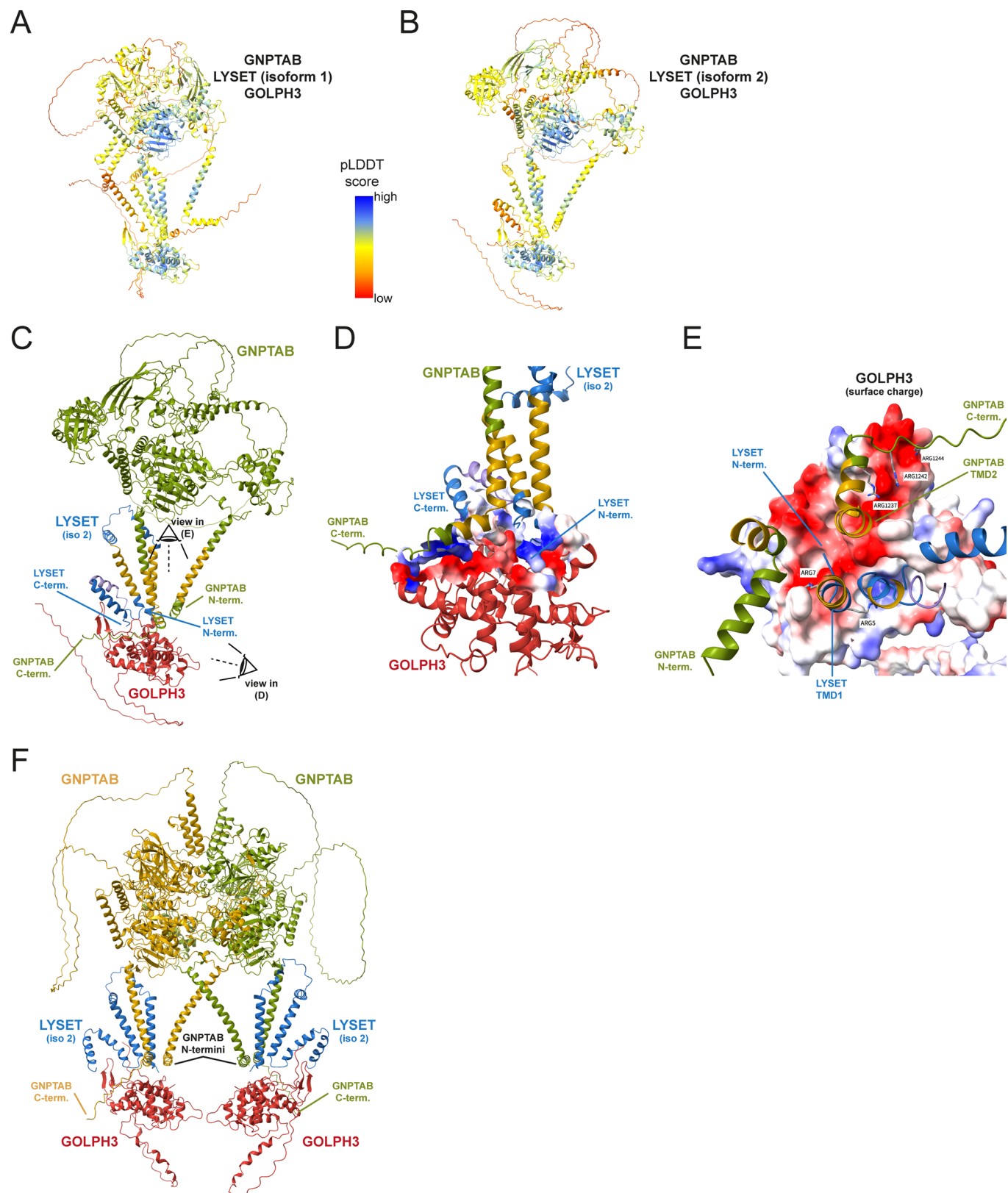

**Figure EV2. AlphaFold 3 predictions of a ternary interaction of GOLPH3, LYSET and GNPTAB.**

(A) Structure of the predicted GOLPH3:LYSET (isoform 1):GNPTAB complex shown in Fig. 6A with coloring reflecting the prediction score. (B) Structure of the predicted GOLPH3:LYSET (isoform 2):GNPTAB complex with prediction score coloring. (C) Structure of the complex shown in (B). Individual subunits are given in different colors. Annotated TMD regions are colored in yellow. In case of LYSET, a putative TMD region predicted by DeepTMHMM (Hallgren et al, 2022) is shown in purple. (D) Side view the complex containing LYSET isoform 2 shown in (B, C), highlighting close proximity of GOLPH3 and the LYSET N-terminus as well as the GNPTAB C-terminus. Surface electrostatic potential is shown for selected amino acid side chains. (E) Same complex as shown in (B, C), but top-down view from the membrane plane onto the GOLPH3 surface. Positively charged amino acid side chains in LYSET and GNPTAB are shown as stick representation. (F) Alphafold 3 prediction of a complex containing a GNPTAB dimer (yellow and green) as well two LYSET (isoform 2) and two GOLPH3 subunits.

