## [Peer Review File · The EMBO Journal]

GOLPH3 and GOLPH3L maintain Golgi localization of LYSET and a functional mannose 6-phosphate transport pathway

Berit Brauer, Zilei Chen, Felix Beirow, Jiaran Li, Daniel Meisinger, Emanuela Capriotti, Michaela Schweizer, Lea Wagner, Jascha Wienberg, Laura Hobohm, Lukas Blume, Wenjie Qiao, Yoshiki Narimatsu, Jan Carette, Henrik Clausen, Dominic Winter, Thomas Braulke, Sabrina Jabs, and Matthias Voss

Corresponding author(s): Matthias Voss (mvoss@biochem.uni-kiel.de) , Sabrina Jabs (s.jabs@ikmb.uni-kiel.de)

Review Timeline:

Submission Date:	6th Jun 24
Editorial Decision:	23rd Jul 24
Revision Received:	5th Oct 24
Editorial Decision:	16th Oct 24
Revision Received:	17th Oct 24
Accepted:	21st Oct 24

Editor: William Teale

Transaction Report:

Dear Matthias,

Thank you again for the submission of your manuscript entitled "GOLPH3/3L-dependent Golgi retention of LYSET ensures M6P-tagging of lysosomal enzymes and B4GALT5" (EMBOJ-2024-118096) and for your patience during an unusually long review process. We have now received the reports from the referees, which I copy below.

As you can see from the comments, referee #2 suggests some further experiments; however, I am not convinced that these fall within the scope of the present study. Referees #1 and #3 raise points that should be discussed, and referees #2 and #3 comment that the manuscript might be rearranged to make it more accessible to a more general audience. These issues will require your attention before your manuscript can be published in The EMBO Journal.

However, based on the overall interest expressed in the reports, I would like to invite you to address the comments of all referees in a revised version of the manuscript. I should add that it is The EMBO Journal policy to allow only a single major round of revision and that it is therefore important to resolve the main concerns at this stage. I believe the concerns of the referees are reasonable and addressable, but please contact me if you have any questions, need further input on the referee comments or if you anticipate any problems in addressing any of their points. A Zoom call to discuss the concerns of the referees might be useful; please suggest a time (after you have had a chance to dissect the reports) that would be convenient. Please, follow the instructions below when preparing your manuscript for resubmission.

I would also like to point out that as a matter of policy, competing manuscripts published during this period will not be taken into consideration in our assessment of the novelty presented by your study ("scooping" protection). We have extended this 'scooping protection policy' beyond the usual 3 month revision timeline to cover the period required for a full revision to address the essential experimental issues. Please contact me if you see a paper with related content published elsewhere to discuss the appropriate course of action.

Again, please contact me at any time during revision if you need any help or have further questions.

Thank you very much again for the opportunity to consider your work for publication. I look forward to your revision.

Best regards,

William

William Teale, Ph.D.
Editor
The EMBO Journal

When submitting your revised manuscript, please carefully review the instructions below and include the following items:

- 1) a .docx formatted version of the manuscript text (including legends for main figures, EV figures and tables). Please make sure that the changes are highlighted to be clearly visible.
- 2) individual production quality figure files as .eps, .tif, .jpg (one file per figure).
- 3) a .docx formatted letter INCLUDING the reviewers' reports and your detailed point-by-point response to their comments. As part of the EMBO Press transparent editorial process, the point-by-point response is part of the Review Process File (RPF), which will be published alongside your paper.
- 4) a complete author checklist, which you can download from our author guidelines ([https://wol-prod-cdn.literatumonline.com/pb-assets/embo-site/Author Checklist%20-%20EMBO%20J-1561436015657.xlsx](https://wol-prod-cdn.literatumonline.com/pb-assets/embo-site/Author%20Checklist%20-%20EMBO%20J-1561436015657.xlsx)). Please insert information in the checklist that is also reflected in the manuscript. The completed author checklist will also be part of the RPF.
- 5) Please note that all corresponding authors are required to supply an ORCID ID for their name upon submission of a revised manuscript.
- 6) We require a 'Data Availability' section after the Materials and Methods. Before submitting your revision, primary datasets

produced in this study need to be deposited in an appropriate public database, and the accession numbers and database listed under 'Data Availability'. Please remember to provide a reviewer password if the datasets are not yet public (see <https://www.embopress.org/page/journal/14602075/authorguide#datadeposition>). If no data deposition in external databases is needed for this paper, please then state in this section: This study includes no data deposited in external repositories. Note that the Data Availability Section is restricted to new primary data that are part of this study.

Note - All links should resolve to a page where the data can be accessed.

8) For data quantification: please specify the name of the statistical test used to generate error bars and P values, the number (n) of independent experiments (specify technical or biological replicates) underlying each data point and the test used to calculate p-values in each figure legend. The figure legends should contain a basic description of n, P and the test applied. Graphs must include a description of the bars and the error bars (s.d., s.e.m.).

9) We would also encourage you to include the source data for figure panels that show essential data. Numerical data can be provided as individual .xls or .csv files (including a tab describing the data). For 'blots' or microscopy, uncropped images should be submitted (using a zip archive or a single pdf per main figure if multiple images need to be supplied for one panel). Additional information on source data and instruction on how to label the files are available at .

10) We replaced Supplementary Information with Expanded View (EV) Figures and Tables that are collapsible/expandable online (see examples in <https://www.embopress.org/doi/10.15252/embj.201695874>). A maximum of 5 EV Figures can be typeset. EV Figures should be cited as 'Figure EV1, Figure EV2" etc. in the text and their respective legends should be included in the main text after the legends of regular figures.

12) Our journal encourages inclusion of *data citations in the reference list* to directly cite datasets that were re-used and obtained from public databases. Data citations in the article text are distinct from normal bibliographical citations and should directly link to the database records from which the data can be accessed. In the main text, data citations are formatted as follows: "Data ref: Smith et al, 2001" or "Data ref: NCBI Sequence Read Archive PRJNA342805, 2017". In the Reference list, data citations must be labeled with "[DATASET]". A data reference must provide the database name, accession number/identifiers and a resolvable link to the landing page from which the data can be accessed at the end of the reference. Further instructions are available at .

We realize that it is difficult to revise to a specific deadline. In the interest of protecting the conceptual advance provided by the work, we recommend a revision within 3 months (21st Oct 2024). Please discuss the revision progress ahead of this time with the editor if you require more time to complete the revisions. Use the link below to submit your revision:

Referee #1:

The Golgi membrane protein LYSET was recently identified as being a binding partner of the GNTAB enzyme that generates the mannose-6-phosphate (M6P) modification that sorts lysosomal hydrolases to the Golgi. This finding attracted a lot of interest given the relevance of M6P to cellular physiology, lysosome function and disease. LYSET plays an important role in the M6P system as its deletion results in loss of Golgi retention of GNTAB and thus loss of M6P addition. However, the mechanism by which LYSET ensures Golgi localisation of GNTAB was unclear.

This manuscript investigates role of LYSET and provides extensive evidence that it binds to the Golgi protein GOLPH3 and its paralogue GOLPH3L, and this interaction serves to retain LYSET and GNTAB in the Golgi. This is unexpected as GOLPH3 has only been reported to bind to the short tails of Golgi enzymes that have only a single TMD, and it has not been previously linked to the M6P system. It is known that many of these Golgi enzymes are cleaved by the Golgi protease SPPL3 when they are not correctly localised and this cleavage means that their luminal domains are secreted. The authors extend their study by finding that at least one Golgi enzyme (B4GALT5) is modified by M6P, and this modification is used to sort the luminal domain to the lysosome following cleavage with SPPL3. This is another unexpected finding and suggests that the M6P system has a role in Golgi quality control in addition to its well-established role in sorting of lysosomal hydrolases.

Overall, the experimental data in the paper is of an exceptionally high quality and fully support the conclusions drawn. The effects observed are very clear, and often demonstrated in multiple cell lines with many controls. The authors often complement analysis of the effects of gene deletions on single proteins with proteomic analysis of whole cell proteomes or secretomes which adds further robustness to the data. Finally, the data figures are clearly labelled, and the findings are summarised with clear and helpful diagrams. Indeed, the technical quality of this study is as high as any paper I have seen in the last few years. There are a few minor points that should be addressed in the presentation and discussion which are listed below. Otherwise, I am very happy to recommend acceptance of what is an outstanding manuscript.

1) The authors present clear evidence that B4GALT5 appears in the medium when LYSET is deleted (Figure 1B and 1C). They should comment on the lack of detectable B4GALT5 in the blot of the membrane lysate which I guess reflects the greater complexity of this sample.

2) In Figure 3A the authors show a volcano plot comparing the secretome from WT HEK293 to those lacking GOLPH3/GOLPH3L. It appears that some Type II Golgi enzymes are secreted less in the mutant than WT (eg B3GNT2 and

GCNT1). Is it possible that SPPL3 is partially mislocalised in the mutant? Or is there some cellular response to losing other enzymes? This should be discussed briefly as it is potentially interesting.

3) Figure 3E shows that LYSET remains stable in cells lacking GOLPH3 but not GOLPH3L. This is in contrast to some other proteins. This might suggest that LYSET is only recognised by GOLPH3L, but they later show that the double deletion can be rescued by GOLPH3. Thus, it is more likely that even low levels of GOLPH3 activity are enough to retain LYSET. This should be briefly discussed.

4) In Figure 3H the authors show a couple of EM images to argue that the Golgi may be slightly disorganised in cells lacking GOLPH3 and GOLPH3L. This is the only piece of data in the paper that is not convincing. The complex shape of the Golgi means that its appearance can vary depending on where an EM thin section is taken, and so conclusions about changes in Golgi morphology require three dimensional data from serial EM or tomography, with analysis applied to multiple stacks. The conclusion is in any case irrelevant to the rest of the paper and so I suggest that this Figure is simply deleted.

5) In Supp Figure 8B and 8C the authors show an AlphaFold2 prediction for the structure of LYSET bound to GOLPH3. I strongly suggest that they include all or part of GNTAB as well. AlphaFold3 confidently predicts that the C-terminal TMD of GNTAB binds to two of the TMDs of LYSET and this brings more basic residues (from GNTAB) into the interface with GOLPH3. This prediction should also be moved to the main figures.

Referee #2:

In this study, the authors have addressed the molecular basis for Golgi retention of Golgi N-acetylglucosamine-1-phosphotransferase (GNPT) retention factor LYSET/TMEM251. Evidence was presented suggesting that GOLPH3 and GOLPH3L interact with positive charged residues of LYSET on the cytoplasmic region on one hand and with COPI on the other hand to mediate its dynamic Golgi retention. In addition, it was shown that B4GALT5 is cleaved by SPPL3 and the cleaved luminal domain is modified by mannose 6-phosphate (M6P) tag for sorting to the lysosome. Therefore, GOLPH3/GOLPH3L is essential for the integrity of the M6P-tagging machinery and homeostasis of lysosomes by maintaining Golgi retention of LYSET. Generally interesting but the manuscript was a bit challenging to read and the abstract should be revised to better reflect the key points in a cohesive way. Some other points are listed below.

1: it will be important to demonstrate mechanistically that COPI interacts with LYSET in a GOLPH3/GOLPH3L-dependent manner or GOLPH3/GOLPH3L, COPI and LYSET dynamically interact in a mutually dependent manner?

2: what is the direct or indirect link of LYSET in SPPL3 dependent cleavage of B4GALT5? and lysosomal sorting?

3: It will be significant if the physiological relevance can be addressed by some in vivo experiments.

Referee #3:

This manuscript follows previous work by Voss and colleagues investigating the secretion of Golgi resident proteins. They have previously shown that a large number of glycan processing enzymes in the Golgi are secreted following cleavage by the protease SPPL3, which liberates the enzymes' ectodomains to allow their constitutive secretion. This study combines their work on SPPL3 cleavage mediated secretion with mechanistic insight linking it to intra-Golgi retrograde sorting via the proteins Golph3/3L. It is revealed that the transmembrane protein LYSET, which has previously been shown to ensure Golgi retention of the GlcNAc phosphotransferase complex (which initiates mannose-6-phosphate (M6P) tagging), is a Golph3/3L client. In the absence of Golph3/3L, LYSET is lost from the Golgi, causing loss of M6P tagging, which in turn promotes secretion of cleaved B4GalT5. This is an interesting paper linking intra-Golgi sorting via the Golph3/3L adaptors and post-Golgi sorting via mannose-6-phosphate (M6P) and proteolytic processing of resident proteins in the Golgi. Prominently, it highlights a new type of Golph3/3L client, adding significant new information to the way we can think about intra-Golgi sorting. I think that the study would be very interesting for readers of EMBO J and have a few recommendations below which may in my opinion improve the clarity of the authors' arguments.

The majority of the paper focusses on investigating the Golph3/3L mediated retention of LYSET in the Golgi, but it starts with the B4GalT5 angle to lead into the LYSET retention story. There is currently a problem with the B4GalT5 part, and it does not gel well with the rest. Two main issues cause this:

1. The B4GalT5 data prompt the authors to propose a model (Fig 6B) by which M6P tagging of cleaved Golgi-enzyme ectodomains prevents their secretion by targeting them to the lysosome. This model is consistent with the data presented, but its support in the presented data is rather weak. It is a speculative model, much more so than 6A or 6C. While this speculation is acknowledged in some parts of the manuscript, this is not done in the description of Fig 6B, and too strongly implied in some other places. I do not think that a further experiment is needed unless this point should be a prominent conclusion in the paper (see also below in point 2).

2. M6P tagging and targeting of non-lysosomal enzymes to the lysosome is an interesting point. I am not a lysosomal expert, and as such have not been aware of such a mechanism. The introduction does not help me in this regard, and it is not until very late in the manuscript where an off-hand comment made me aware that this mechanism, which I thought was really cool and should really be the main selling point of the paper, is not that new. The interplay between this point and point 1 detracts from the main selling point of the paper. I would introduce M6P tagging of non-lysosomal proteins better and tone down Fig 6B to ensure that this aspect really does what I think may have been the original intention, that is to introduce LYSET as a new Golph3/3L client.

3. If the B4GalT5 M6P tagging storyline is reduced in prominence, the main new finding is that LYSET is a new type of Golph3/3L client. There is a very good amount of data to show that LYSET is indeed sorted in a Golph3/3L dependent way, but it being a direct client hinges on the bioinformatic analysis revealing a basic patch in the cytoplasmic region of LYSET. I feel that some experimental verification of this patch or some other more direct evidence of LYSET-Golph3/3L binding could be nice to provide robust evidence for the discovery of this novel class of Golph3/3L client.

Other points

1. Kuhn et al 2015 identified B4GalT5 among the proteins depleted from the secretome of SPPL3 KO cells. These cells are LYSET positive, so should have re-routed cleaved B4GalT5 to the lysosome. What was different in those cells?

2. It is hard to tell what band is B4GalT5 in the whole cell lysate samples, and there is a weak band that is likely non-specific in the WT secreted sample in Fig 1B/C where this protein is probed for. Some clarity would be good.

3. In Fig 3A SPPL3 substrates can be found in areas other than the region upregulated upon Golph deletion. How is this explained? Is it possible that the change in secretion +/- Golph deletion does not provide much mechanistic information because there are several different possible mechanisms to be reckoned with?

4. In Fig 3D I think it is misleading to include peripheral membrane proteins. Most of these are not sorted via vesicular trafficking.

5. In Fig 4 most cathepsins behave as expected, but for Cat D there is no decrease in the level of its mature form in the cell while the immature form is secreted in Golph KO cells. Why?

6. In contrast to what the text states, I cannot see GNPTa to increase in the lysosome fraction of KO HEK cells upon protease inhibitor treatment in Fig 5.

7. Please include black and white single channel images for all fluorescent image panels, not just composites. It is difficult to see the implied colocalization in many of the images.

Point-by-point response – manuscript EMBOJ-2024-118096

We thank the referees of our manuscript for positive comments and insightful suggestions. Based on these thoughtful suggestions from the referees we have included a substantial amount of new data. The revised manuscript is now expanded and re-organized into seven main figures, two Extended View figures, and ten Appendix figures.

Please find below the reviewers' comments in full with our responses in black.

Referee #1:

[...]

Overall, the experimental data in the paper is of an exceptionally high quality and fully support the conclusions drawn. The effects observed are very clear, and often demonstrated in multiple cell lines with many controls. The authors often complement analysis of the effects of gene deletions on single proteins with proteomic analysis of whole cell proteomes or secretomes which adds further robustness to the data. Finally, the data figures are clearly labelled, and the findings are summarised with clear and helpful diagrams. Indeed, the technical quality of this study is as high as any paper I have seen in the last few years. There are a few minor points that should be addressed in the presentation and discussion which are listed below. Otherwise, I am very happy to recommend acceptance of what is an outstanding manuscript.

We would like to thank the reviewer for taking the time to thoroughly examine our manuscript and are delighted to read about the positive assessment of our study and its quality. We also thank the reviewer for the constructive input and suggestions which we have addressed in the revised version of the manuscript as detailed in the following.

1) The authors present clear evidence that B4GALT5 appears in the medium when LYSET is deleted (Figure 1B and 1C). They should comment on the lack of detectable B4GALT5 in the blot of the membrane lysate which I guess reflects the greater complexity of this sample.

(a similar point was raised by referee #3, see below)

Indeed, using mAbs 6D4 and 1E10, yet also commercial B4GALT5-specific antibodies (not shown), we cannot detect cell-endogenous B4GALT5 in whole-cell or membrane lysates by immunoblot. This may well reflect the greater complexity of the samples as indicated by the reviewer. Furthermore, low sensitivity of the antibody and low steady state expression levels of the glycosyltransferase might additionally complicate the detection of the endogenous protein. The latter is also reflected by the fact, that we did not detect endogenous B4GALT5 in our proteomics data obtained from whole cell lysates (Figure 3D). B4GALT5 overexpressed from a strong viral promoter can principally be detected in cell lysates using antibodies used in this study (not shown). We have included a statement discussing steady state expression levels and low antibody sensitivity on **page 6** of the revised manuscript.

In addition, to enable detection of near-endogenous B4GALT5 intracellularly, we have now included a new experimental panel (**Figure 2G**) for which we implemented a B4GALT5-mCherry RUSH reporter. Following biotin treatment-induced release from the ER, the reporter (detected by both mCherry- and B4GALT5-specific antibodies) is released into the media of LYSET KO yet not WT cells. As expected, the reporter is size-shifted compared to cell-endogenous B4GALT5 but compared to the amount of cell-endogenous B4GALT5 co-secreted the B4GALT5 overexpression achieved with this system is moderate. Using the mCherry antibody, this system enabled the visualization of B4GALT5 in cell lysates and revealed differences in B4GALT5 maturation in LYSET KO and WT cells, which are well in line with impaired M6P tagging of the B4GALT5 reporter in LYSET KO cells.

2) In Figure 3A the authors show a volcano plot comparing the secretome from WT HEK293 to those lacking GOLPH3/GOLPH3L. It appears that some Type II Golgi enzymes are secreted less in the mutant than WT (eg B3GNT2 and GCNT1). Is it possible that SPPL3 is partially mislocalised in the mutant? Or is there some cellular response to losing other enzymes? This should be discussed briefly as it is potentially interesting.

(a similar point was also raised by referee #3)

The reviewer (along with reviewer #3) correctly noted that our secretome analyses of GOLPH3/GOLPH3L-deficient cells not only revealed the discussed hypersecretion of B4GALT5 but also that secretion of other type II membrane proteins/Golgi enzymes is profoundly altered. Apart from B4GALT5, several other enzymes (e.g. MAN2A1, MGAT2) appear to be hypersecreted, while other established SPPL3 substrates (e.g. CANT1, MGAT5) are secreted less efficiently from GOLPH3/GOLPH3L KO cells. We feel that a dissection of this complex phenotype is indeed warranted but would be beyond the scope of the current manuscript which already has a complex and multi-layered storyline.

Indeed, a possible (partial) explanation for such observations could be a GOLPH3-dependent altered localization or activity of SPPL3. We previously reported that loss of GOLPH3/GOLPH3L in cells with endogenously HA-tagged SPPL3 is not accompanied by overt differences in SPPL3 distribution in nocodazole-induced Golgi mini-stacks (Truberg et al. (2022), BBA Mol Cell Res).

Alternatively, GOLPH3/GOLPH3L-deficient cells undoubtedly display a glycosylation phenotype due to Golgi enzyme redistribution/destabilization and in light of the recent observations by Hirata et al. (2022), *Commun Biol*, altered glycosylation patterns on substrates may affect their SPPL3-mediated intramembrane proteolysis.

3) Figure 3E shows that LYSET remains stable in cells lacking GOLPH3 but not GOLPH3L. This is in contrast to some other proteins. This might suggest that LYSET is only recognised by GOLPH3L, but they later show that the double deletion can be rescued by GOLPH3. Thus, it is more likely that even low levels of GOLPH3 activity are enough to retain LYSET. This should be briefly discussed.

As the reviewer indicates, there is indeed evidence in the literature that certain Golgi enzymes/GOLPH3 clients are already destabilized/mislocalized upon knockdown/knockout of only GOLPH3, i.e. remaining cellular GOLPH3L levels are likely insufficient to sustain Golgi enzyme localization. However, using our KO cell lines, we find that loss of GOLPH3 alone does not lead to a marked destabilization of endogenous clients such as GALNT7, CANT1, GPP130. This discrepancy with work of others could stem from their use of ectopically expressed Golgi enzyme reporters that may require more GOLPH3/GOLPH3L to sustain retrieval. With that in mind, LYSET in fact behaves very similar to other cell-endogenous GOLPH3/GOLPH3L clients. We have included additional Appendix Figure panels (**Appendix Figure S3D and S5C**) to show that levels of established GOLPH3/GOLPH3L clients in GOLPH3 KO cells are not destabilized.

Moreover, we have selectively targeted *GOLPH3L* in HEK293 cells to establish cells that only express GOLPH3 and included these data in the revised version of the manuscript. In these cells, endogenous levels of neither established clients nor LYSET are markedly reduced (**Appendix Figure S3G**). Collectively, this further supports the notion that GOLPH3 and GOLPH3L are functionally redundant and that low-level expression of either GOLPH3 or GOLPH3L is required and sufficient to retain type II membrane proteins and LYSET in the Golgi apparatus. As requested by the reviewer, this is now also specifically pointed out in the discussion (**page 14**).

4) In Figure 3H the authors show a couple of EM images to argue that the Golgi may be slightly disorganised in cells lacking GOLPH3 and GOLPH3L. This is the only piece of data in the paper that is not convincing. The complex shape of the Golgi means that its appearance can vary depending on where an EM thin section is taken, and so conclusions about changes in Golgi morphology require three dimensional data from serial EM or tomography, with analysis applied to multiple stacks. The conclusion is in any case irrelevant to the rest of the paper and so I suggest that this Figure is simply deleted.

To our knowledge, apart from previous work by Welch et al (*J Cell Biol*, 2021), this manuscript represents only the second study that reports on GOLPH3/GOLPH3L double-deficient cell lines. In the aforementioned study no notable Golgi abnormalities have been reported and we feel that the first ultrastructural description of the affected organelle should be part of this study and can be of relevance to an interested readership.

However, to address the reviewer's concern with single-section EM images, we have extended our dataset to n = 5, have removed the panels from the main figure and have now placed them in **Appendix Figure S6**.

5) In Supp Figure 8B and 8C the authors show an AlphaFold2 prediction for the structure of LYSET bound to GOLPH3. I strongly suggest that they include all or part of GNPTAB as well. AlphaFold3 confidently predicts that the C-terminal TMD of GNPTAB binds to two of the TMDs of LYSET and this brings more basic residues (from GNPTAB) into the interface with GOLPH3. This prediction should also be moved to the main figures.

We thank the reviewer for this valuable suggestion. With AlphaFold 3 now available, we have performed the modelling of a GOLPH3:LYSET:GNPTAB complex as suggested by the reviewer. Indeed, this predicted that the GNPTAB C-terminal region engages in interactions with LYSET and GOLPH3 as already observed by the reviewer. We have further extended this by also modelling a complex including a GNPTAB dimer. In sum all these predictions further support the interaction of LYSET and GOLPH3 but also indicate that GNPTAB may similarly be engaged in the formation of a complex. However, the activation of the GNPTAB precursor by Site-1-protease-mediated cleavage into mature alpha- and beta-subunits is not included in the prediction and might additionally affect the structure of the heteromeric complex.

As requested, we have included the AlphaFold 3 prediction for LYSET isoform 1 in a main manuscript figure (**Fig. 6G-I**) and have compiled further structural predictions in a novel **Expanded View Figure EV2**.

Referee #2:

In this study, the authors have addressed the molecular basis for Golgi retention of Golgi N-acetylglucosamine-1-phosphotransferase (GNPT) retention factor LYSET/TMEM251. Evidence was presented suggesting that GOLPH3 and GOLPH3L interact with positive charged residues of LYSET on the cytoplasmic region on one hand and with COPI on the other hand to mediate its dynamic Golgi retention. In addition, it was shown that B4GALT5 is cleaved by SPPL3 and the cleaved luminal domain is modified by mannose 6-phosphate (M6P) tag for sorting to the lysosome. Therefore, GOLPH3/GOLPH3L is essential for the integrity of the M6P-tagging machinery and homeostasis of lysosomes by maintaining Golgi retention of LYSET. Generally interesting but the manuscript was a bit challenging to read and the abstract should be revised to better reflect the key points in a cohesive way. Some other points are listed below.

We thank the reviewer for this comment and have edited the manuscript text and abstract to hopefully facilitate the reading and emphasize the key findings.

1: it will be important to demonstrate mechanistically that COPI interacts with LYSET in a GOLPH3/GOLPH3L-dependent manner or GOLPH3/GOLPH3L, COPI and LYSET dynamically interact in a mutually dependent manner?

The reviewer is right that experimental evidence of dynamic interactions between GOLPH3/GOLPH3L, COPI and LYSET and between COPI and LYSET in a GOLPH3/GOLPH3L-dependent manner are important issues. However, GNPTAB has also been shown to directly bind COPI via its N-terminus (Liu et al. 2018). In addition, the AlphaFold 3 model that we have now included in response to Reviewer 1 (**Fig. 6G-I, Fig. EV2**) further suggests also the C-terminus of GNPTAB may provide an interaction surface with the GOLPH3 adaptor. This suggests that the retrieval of the heteromeric complex is warranted for by different partially redundant mechanisms. Since LYSET, however, is required for the stability of GNPTAB, the influence of direct or GOLPH3-mediated COPI-binding to GNPTAB and the GOLPH3-interaction with LYSET are difficult to dissect and beyond the scope of the present study. We have extended our discussion of this increased complexity in the results part (**page 12**) and in the discussion (**page 14/15**).

2: what is the direct or indirect link of LYSET in SPPL3 dependent cleavage of B4GALT5? and lysosomal sorting?

Our data do not support that LYSET has a direct effect on SPPL3 as intramembrane cleavage and secretion of the SPPL3 substrates tested (MGAT5, B4GALT1, B4GAT1, EXTL3, GALNT2; Figs. 1B and C) is not altered in LYSET KO cells. LYSET (but also GNPT) deficiency appears to rather selectively cause a B4GALT5 hypersecretion. We have and report no evidence that SPPL3 activity towards B4GALT5 is elevated under these conditions. We unambiguously demonstrate that soluble B4GALT5, the cleavage product, is secreted in LYSET- or GNPTAB-deficient cells that lack functional M6P tagging. In LYSET-proficient cells, B4GALT5 is subject to M6P tagging and thus trafficked to lysosomes and not secreted in substantial amounts following SPPL3-mediated cleavage.

We apologize that this had not been made clear enough in the submitted version of the manuscript and adjusted the text in the revised version of the manuscript to make this clearer (**page 6**).

3: It will be significant if the physiological relevance can be addressed by some in vivo experiments.

Undoubtedly, work using GOLPH3/GOLPH3L-double deficient animal models would promise important insights into the in vivo relevance of our observations reported in the manuscript. However, given that neither GOLPH3- nor GOLPH3L-deficient and in particular no GOLPH3/GOLPH3L double-deficient animals (which would be strictly required based on our present data) are established/available in any of our laboratories, it is not feasible and far beyond the current scope to further complement our study with in vivo experiments.

Referee #3:

[...]This is an interesting paper linking intra-Golgi sorting via the Golph3/3L adaptors and post-Golgi sorting via mannose-6-phosphate (M6P) and proteolytic processing of resident proteins in the Golgi. Prominently, it highlights a new type of Golph3/3L client, adding significant new information to the way we can think about intra-Golgi sorting. I think that the study would be very interesting for readers of EMBO J and have a few recommendations below which may in my opinion improve the clarity of the authors' arguments.

We thank the reviewer for the in-depth assessment of our study, the appreciative feedback and valuable suggestions, which we have addressed now as detailed below.

The majority of the paper focusses on investigating the Golph3/3L mediated retention of LYSET in the Golgi, but it starts with the B4GalT5 angle to lead into the LYSET retention story. There is currently a problem with the B4GalT5 part, and it does not gel well with the rest. Two main issues cause this:

1. The B4GalT5 data prompt the authors to propose a model (Fig 6B) by which M6P tagging of cleaved Golgi-enzyme ectodomains prevents their secretion by targeting them to the lysosome. This model is consistent with the data presented, but its support in the presented data is rather weak. It is a speculative model, much more so than 6A or 6C. While this speculation is acknowledged in some parts of the manuscript, this is not done in the description of Fig 6B, and too strongly implied in some other places. I do not think that a further experiment is needed unless this point should be a prominent conclusion in the paper (see also below in point 2).

To eliminate this issue raised by the reviewer, we have removed the admittedly speculative panel B from formerly Fig. 6 in the revised version of the manuscript. Also, the corresponding generalizing statement referring to this panel in the discussion was removed.

2. M6P tagging and targeting of non-lysosomal enzymes to the lysosome is an interesting point. I am not a lysosomal expert, and as such have not been aware of such a mechanism. The introduction does not help me in this regard, and it is not until very late in the manuscript where an off-hand comment made me aware that this mechanism, which I thought was really cool and should really be the main selling point of the paper, is not that new. The interplay between this point and point 1 detracts from the main selling point of the paper. I would introduce M6P tagging of non-lysosomal proteins better and tone down Fig 6B to ensure that this aspect really does what I think may have been the original intention, that is to introduce LYSET as a new Golph3/3L client.

We respectfully disagree with the reviewer that the M6P tagging of non-lysosomal membrane proteins is not a main selling point of our manuscript. The reports we cite that have previously detected M6P-residues on membrane proteins are exclusively mass spectrometric studies that neither comment extensively on the non-lysosomal membrane proteins found in their datasets nor do they thoroughly and formally corroborate this M6P-tagging of these proteins experimentally.

These studies differ fundamentally from our present data identifying a novel mechanism for selective M6P-modification of B4GALT5. This became apparent only through the analysis of LYSET KO secretomes, and the link to the secondary loss of GNPT. The re-analysis of previously published M6P-dependent proteome data then revealed that M6P-tagged B4GALT5 may represent a prototype for M6P-containing type I and II transmembrane proteins. The biological significance of the M6P modification for the functions and homeostasis of the Golgi, lysosomes or other compartments will be subject to future studies.

3. If the B4GalT5 M6P tagging storyline is reduced in prominence, the main new finding is that LYSET is a new type of Golph3/3L client. There is a very good amount of data to show that LYSET is indeed sorted in a Golph3/3L dependent way, but it being a direct client hinges on the bioinformatic analysis revealing a basic patch in the cytoplasmic region of LYSET. I feel that some experimental verification of this patch or some other more direct evidence of LYSET-Golph3/3L binding could be nice to provide robust evidence for the discovery of this novel class of Golph3/3L client.

To experimentally support that the basic amino acids found in the LYSET N-terminal region are of relevance, we have established expression constructs for a panel of LYSET point mutations and have assessed stability of the LYSET mutants. Our findings in transfected LYSET KO and WT HEK293 cells (**Fig. EV1G and Fig. 6C**) indeed suggest that in particular R37 is important for LYSET stability as expression levels of the R37A variant are severely reduced. As we noted the most drastic reduction for the R37A/R39A double-mutant and wanted to eliminate overexpression-related issues, we further established LYSET R37A/R39A knock-in cell lines (**Fig. 6D to F**) which feature strongly reduced but not entirely absent cellular LYSET levels, which can only partially sustain maturation of lysosomal enzymes. Given our novel AlphaFold 3 predictions (**Fig. 6G-I and Fig. EV2**) that support the formation of a multimeric complex of GOLPH3, LYSET and the GNPT subunits, with GOLPH3 interfacing not only with LYSET but possibly also with the C-terminus of GNPT β -subunit, a more thorough dissection of the binding modalities of these interaction partners is needed but would require extensive mutagenesis of not only LYSET but also GNPTAB. We feel that such

analyses are not feasible given the time frame granted for the revision of our manuscript and would indeed warrant a separate molecular study.

Other points

1. Kuhn et al 2015 identified B4GalT5 among the proteins depleted from the secretome of SPPL3 KO cells. These cells are LYSET positive, so should have re-routed cleaved B4GalT5 to the lysosome. What was different in those cells?

We thank the reviewer for being this observant and raising this crucial point. Indeed, quantifying metabolically labelled and selectively enriched secreted proteins from murine embryonic fibroblasts, Kuhn et al. previously reported a SPPL3-dependent release of B4GALT5 into cell culture supernatants. We have added a sentence referencing this observation to the discussion (**page 14**).

Indeed, the SPPL3-dependent secretion of B4GALT5 was observed by Kuhn et al. in LYSET-proficient cells. It may well be that the mass spectrometric detection of B4GALT5 is more sensitive than our immunoblotting with antibodies (that also evidently have limitations in regard to their capacity to detect intracellular B4GALT5) and that we have therefore not observed SPPL3-dependent B4GALT5 release from human cell lines with intact M6P tagging. Our mass spectrometric data from murine embryonic fibroblasts (Fig. 1A, Richards et al. 2022) show that there is a significant increase of B4GALT5 secretion from LYSET-deficient cells compared to WT cells. This does not imply that secretion from WT cells is completely absent, and indeed WT cells globally release low levels of immature lysosomal hydrolases (e.g. Hex B, various cathepsins in Figure 4) that are being captured by mannose 6-phosphate receptors at the plasma membrane, which may equally be the case for B4GALT5. In line with this, expression of the B4GALT5-mCherry RUSH reporter (new **Fig. 2G**) led to observe low-level release of B4GALT5-mCherry also from LYSET-proficient cells.

2. It is hard to tell what band is B4GalT5 in the whole cell lysate samples, and there is a weak band that is likely non-specific in the WT secreted sample in Fig 1B/C where this protein is probed for. Some clarity would be good.

(a similar point was raised by referee #1, please also see comment above)

The unspecific bands in conditioned medium samples have now been labelled with an asterisks in **Figs. 1B, 1C**, but also **2C, 2D** and **2E**.

Explaining the reviewer's concern about B4GALT5 in cell or membrane lysate samples, we feel that in the panels shown neither B4GALT5 mAb 6D4 nor 1E10 robustly and reliably detected endogenous B4GALT5 in immunoblots. To circumvent this issue and nonetheless study maturation of B4GALT5 in cells, we have established a B4GALT5-mCherry RUSH reporter (**new panel G in Figure 2**). Using this and an mCherry-specific antibody, we can visualize B4GALT5 in cell lysates and observed different maturation patterns in WT and LYSET KO cells, which are in line with impaired M6P-tagging in the former.

3. In Fig 3A SPPL3 substrates can be found in areas other than the region upregulated upon Golph deletion. How is this explained? Is it possible that the change in secretion +/- Golph deletion does not provide much mechanistic information because there are several different possible mechanisms to be reckoned with?

(a similar point was raised by referee #1)

Indeed, the changes in Golgi enzyme/type II membrane protein secretion evident from the secretome data presented in Fig. 3A appear to be rather complex, with B4GALT5 being hypersecreted (as expected from data presented in Figs. 1 & 2 due to the impaired M6P tagging in GOLPH3/GOLPH3L KO cells) but other type II proteins and known GOLPH3/GOLPH3L clients (Welch et al. (2021), J Cell Biol) being secreted less.

We strongly believe that this phenotype is interesting and warrants in-depth dissection but we feel that this would be beyond the scope of this manuscript. It is, however, as the reviewer suggests, a valid possibility that this phenotype is due to the interplay of multiple mechanisms in GOLPH3/GOLPH3L deficient cells.

4. In Fig 3D I think it is misleading to include peripheral membrane proteins. Most of these are not sorted via vesicular trafficking.

The intention of Fig. 3D is to further corroborate our immunoblot data from panel 3C that the abundance of Golgi membrane proteins is not globally altered upon depletion of GOLPH3/GOLPH3L. Obviously, if the dataset included type II membrane proteins this would be somewhat skewed since GOLPH3/GOLPH3L is an established trafficking factor for type II membrane protein and thus changes affecting cellular Golgi type II membrane proteins levels are expected (see e.g. Welch et al. (2021), J Cell Biol). For that reason, we choose to specifically filter for Golgi integral and peripheral membrane proteins, excluding type II membrane proteins. In regard to the cellular abundance of Golgi membrane proteins (apart from type II membrane proteins which are to be known maintained in a GOLPH3/GOLPH3L-dependent fashion), we feel that abundance of both non-type II

integral but also peripheral Golgi membrane proteins can be informative, irrespective of their different modes of trafficking/recruitment to the Golgi.

5. In Fig 4 most cathepsins behave as expected, but for Cat D there is no decrease in the level of its mature form in the cell while the immature form is secreted in Golph KO cells. Why?

We have optimized our cathepsin D detection and have specifically looked again at cathepsin D levels and maturation in GOLPH3/GOLPH3L KO HAP1 and HEK293 cells. Immature cathepsin D is evidently released in larger amounts from GOLPH3/GOLPH3L KO cells and displays an impaired maturation in these cells. There is also a notable reduction of mature cathepsin D, but it is indeed and in line with the reviewer's comment not as drastically reduced as the other soluble lysosomal enzymes probed by immunoblot. This may be related to a previously reported M6P-independent sorting of cathepsin D in cell lines (Rijnboutt et al. (1991), J Biol Chem; Markmann et al. (2015), Traffic). The new cathepsin D panels now form a new subpanel of **Appendix Figure S7** and **Fig. 4B** has also been updated.

6. In contrast to what the text states, I cannot see GNPT α to increase in the lysosome fraction of KO HEK cells upon protease inhibitor treatment in Fig 5.

We agree with the reviewer that the stabilization of GNPT α in HEK cells is far less pronounced than the stabilization of GNPT α in HAP1 cells or the stabilization of LYSET in either cell line. This might be explained by different residual protease activities in the different cell lines for which different protease inhibitors might have been required (Pechincha et al. 2022). We have toned down our statement for GNPT α in HEK cells and included densitometric quantification of the effects of the protease inhibitors of three independent experiments (**Fig. 5 E, F**).

7. Please include black and white single channel images for all fluorescent image panels, not just composites. It is difficult to see the implied colocalization in many of the images.

We have included the single panels for all channels (except the DNA-staining) in either the Main Figures (**Fig. 3G, H**) and the new **Appendix Figure S9** (for the composite images shown in Fig. 5).

Dear Matthias,

Thank you for submitting the revised version of your manuscript, which addresses the concerns of the referees. This revised version has now been re-reviewed; I attach the second referee reports to the bottom of this mail. As you will see, you have addressed the referees' concerns to their satisfaction. Reviewer 1 makes some final constructive suggestions which I would like you to consider carefully. Before I can finally accept the manuscript, there are some remaining editorial points which need to be addressed. In this regard, would you please:

- acknowledge funding from the Chinese Scholarship Council and the Faculty of Medicine at Kiel University in our online submission system and from Christian-Albrechts-Universität zu Kiel (CAU) in the manuscript text,
- reduce the number of keywords to five,
- remove the author credit section from the manuscript,
- specify section "Sample definition and in-laboratory replication" since the response is "Yes",
- rename dataset 'Dataset EV1' and update source file names, titles, legends and ms callouts to Dataset EV1; the legend should be uploaded (as is) as a separate tab/sheet in the Excel file,
- add page numbers to the table of contents,
- remove the reagents and tools from the manuscript file and upload as an individual file using the template,
- remove reviewer access codes from dataset PXD052083 and allow full public access, providing full URLs,
- indicate the statistical test used for data analysis in the legends of figures 1A, 3A, D; 4C, D,
- define the nature of n in the legends of figures 1A, 3D, 4C, D, 5E, F,
- define asterisk in the legends of figures 1B, C; 2C,
- define black and red arrowheads in the legend of figure 2G,
- define black arrowheads in the legend of figure 4E,
- define black, grey and open type arrowheads in the legend of figure 6E, and
- remove the "Abbreviations" section from the manuscript,

I look forward to receiving these changes.

EMBO Press is an editorially independent publishing platform for the development of EMBO scientific publications.

Best wishes,

William

William Teale, PhD
Editor
The EMBO Journal
w.teale@embojournal.org

We realize that it is difficult to revise to a specific deadline. In the interest of protecting the conceptual advance provided by the work, we recommend a revision within 3 months (14th Jan 2025). Please discuss the revision progress ahead of this time with the editor if you require more time to complete the revisions. Use the link below to submit your revision:

Referee #1:

The original version of this interesting manuscript was already of a very high quality and the authors have done an excellent job of engaging with my comments and suggestions. For what it is worth, I feel that they have also successfully addressed the comments made by the other reviewers.

I have one very minor suggestion, which is that the authors could add a brief comment in the Discussion as to why this M6P modification is added to B4GALT5 and not the many other Golgi enzymes that are clipped by SPPL3. It suggests that for some reason it would be better to prevent the release of B4GALT5 and rather degrade it, but why released B4GALT5 might be something to avoid is an interesting question that could be at least mentioned.

Apart from this minor point, I am very happy to recommend acceptance of the excellent paper.

Referee #2:

The authors address my points by discussions without additional experiments...for physiological relevance, the authors argued that knockouts are not available. At least the authors should test the physiological relevance of this trafficking regulation at the cellular levels

Referee #3:

The revised manuscript and the accompanying rebuttal address the concerns I raised with the first submission.

Point-by-point response – manuscript EMBOJ-2024-118096R

We would like to thank the reviewers for their time and effort spent on evaluating the revised version of our manuscript.

Enclosed you will find the reviewers' comments in full with our responses in black.

Referee #1:

The original version of this interesting manuscript was already of a very high quality and the authors have done an excellent job of engaging with my comments and suggestions. For what it is worth, I feel that they have also successfully addressed the comments made by the other reviewers.

I have one very minor suggestion, which is that the authors could add a brief comment in the Discussion as to why this M6P modification is added to B4GALT5 and not the many other Golgi enzymes that are clipped by SPPL3. It suggests that for some reason it would be better to prevent the release of B4GALT5 and rather degrade it, but why released B4GALT5 might be something to avoid is an interesting question that could be at least mentioned.

Apart from this minor point, I am very happy to recommend acceptance of the excellent paper.

We are grateful to the reviewer for so thoroughly assessing our initial manuscript and now again the revised version. The constructive feedback has undoubtedly improved our manuscript.

The reviewer again raises an interesting and valid point. Given that a (well conceivable) physiological function of Golgi enzymes cleaved by SPPL3 in the extracellular space currently remains enigmatic, our findings that secretion of substantial amounts of B4GALT5 is in fact prevented by M6P tagging suggest that there might be a benefit of preventing secretion of B4GALT5. We have now added a statement to the discussion that brings this interesting aspect to the readers' attention.

Referee #2:

The authors address my points by discussions without additional experiments...for physiological relevance, the authors argued that knockouts are not available. At least the authors should test the physiological relevance of this trafficking regulation at the cellular levels

Our study clearly demonstrates that LYSET function (i.e. stabilization and Golgi retention of the GNPT complex) is compromised in GOLPH3/GOLPH3L-deficient cells. On top of that, our study establishes that as a further consequence M6P tagging as well as lysosomal enzyme trafficking and maturation is impaired. Hence, we feel that our study already sufficiently establishes the physiological significance of GOLPH3/GOLPH3L- and thus likely COPI-dependent retrieval of LYSET on a cellular level.

Referee #3:

The revised manuscript and the accompanying rebuttal address the concerns I raised with the first submission.

We again thank the reviewer for the constructive and in-depth feedback on our manuscript and are delighted to read that all concerns have been addressed to the reviewer's satisfaction.

Dear Matthias,

I am pleased to inform you that your manuscript has been accepted for publication in the EMBO Journal.

Congratulations on publishing a really impressive study!

Best wishes,

William

William Teale, PhD
Editor
The EMBO Journal
w.teale@embojournal.org
